



# An oxidation flow reactor for simulating and accelerating secondary aerosol formation in aerosol liquid water and cloud droplets

Ningjin Xu[1,2], Chen Le[1,2], David R. Cocker[1,2], Don R. Collins[*1,2]

[1]Department of Chemical and Environmental Engineering, University of California Riverside,  Riverside, CA 92521

[2]College of Engineering - Center for Environmental Research and Technology (CE-CERT), University of California Riverside, Riverside, CA 92507

*Correspondence to*: Don R. Collins (donc@ucr.edu)

## Abstract

Liquid water in cloud droplets and aqueous aerosols serves as an important reaction medium for the formation of secondary aerosol through aqueous-phase reactions (aqSA). Large uncertainties remain in estimates of the production and chemical evolution of aqSA in the dilute solutions found in cloud droplets and the concentrated solutions found in aerosol liquid water, which is partly due to the lack of available measurement tools and techniques. A new oxidation flow reactor (OFR), the Accelerated Production and Processing of Aerosols (APPA) reactor, was developed to measure secondary aerosol formed through gas- and aqueous-phase reactions, both for laboratory gas mixtures containing one or more precursors and for ambient air. For simulating in-cloud processes, droplets formed on monodisperse seed particles are introduced into the top of the reactor and the relative humidity (RH) inside it is controlled to 100 %. Similar measurements made with the RH in the reactor <100 % provide contrasts for aerosol formation with no liquid water and with varying amounts of aerosol liquid water.

The reactor was characterized through a series of experiments and used to form secondary aerosol from known concentrations of an organic precursor and from ambient air. The transmission efficiency of $O_3$ and $CO_2$ for all RH and of $SO_2$ for low RH exceeds 90 %, while it falls to about 70 % for $SO_2$ at 100 % RH. Particle transmission efficiency increases with increasing particle diameter from 0.67 for 0.050 μm particles to 0.98 at 0.20 μm, while that of the ~3.3 μm droplets formed on seed particles is greater than 80 %. The residence time distributions of both gases and particles are narrow relative to other OFRs and lack the tails at long residence time expected with laminar flow. Initial cloud processing experiments focused on



the well-studied oxidation of dissolved $SO_2$ by $O_3$, with observed growth of seed particles resulting from the added sulfuric acid agreeing well with estimates based on the relevant set of aqueous phase reactions. The OH exposure ($OH_{exp}$) for low RH, high RH, and in-cloud conditions was determined experimentally from the loss of $SO_2$ and benzene, and simulated from

the KinSim chemical kinetics solver with inputs of measured 254 nm UV intensity profile through the reactor and loss of $O_3$ due to photolysis. The aerosol yield for benzene at high $OH_{exp}$ ranged from 18 % at low RH with dry seed particles present in the reactor to 59 % with cloud droplets present. Measurement of the composition of the secondary aerosol formed from ambient air using an aerosol mass spectrometer showed that the oxygen to carbon ratio (O:C) of the

organic component increased with increasing RH (and liquid water content).



# 1 Introduction

Atmospheric aerosols have significant roles in air quality and climate (Akimoto, 2003; Seinfeld and Pandis, 2016; Shiraiwa et al., 2017). They consist of organic and inorganic

compounds, with the organic component (organic aerosol; OA) being a substantial contributor to submicron aerosols, accounting for 20 ~ 90 % of aerosol mass loadings worldwide (Kanakidou et al., 2005; Jimenez et al., 2009; Knopf et al., 2018). Aerosol particles are both emitted directly into the atmosphere (primary aerosol) and produced in the atmosphere from reactions involving precursor gases (secondary aerosol) (Canagaratna et al., 2007; Andreae and Rosenfeld, 2008;

Myhre et al., 2013). Secondary aerosol species include inorganic nitrate ($NO_3^-$), sulfate ($SO_4^{2-}$), and ammonium ($NH_4^+$), as well as thousands of organic compounds (secondary organic aerosol; SOA) that, collectively, account for a significant fraction of OA mass (Salcedo et al., 2006; Docherty et al., 2008; Froyd et al., 2009; Hallquist et al., 2009; Ehn et al., 2014). Considerable progress has been made towards understanding the efficiency with which secondary aerosol

forms from gas-phase oxidation of important anthropogenic and biogenic precursors (Shrivastava et al., 2017; Schroder et al., 2018; Bianchi et al., 2019). Much less is known about the production rate and properties of secondary aerosol formed through aqueous-phase reactions in atmospheric liquid water (aqSA). Despite differences of orders of magnitude in liquid water content (LWC), both cloud droplets and aqueous aerosol particles serve as important reaction mediums for the

formation and evolution of aqSA. Experimental and modeling efforts to determine the contribution of aqSA to the total secondary aerosol burden are complicated by the diversity of organic and inorganic precursor gases, the complexity of the chemical pathways and products formed in liquid water, and uncertainties in quantities such as the concentration, composition, and size distribution of droplets. Nevertheless, laboratory and modeling studies have revealed

that the contribution of aqueous reactions of dissolved inorganics and organics to secondary aerosol formation is significant (Lim et al., 2005; Carlton et al., 2006; Carlton et al., 2007; Volkamer et al., 2009; McNeill et al., 2012; Budisulistiorini et al., 2017; Ma et al., 2021; Wang et al., 2021).

Though sulfate formation from aqueous-phase oxidation of sulfur dioxide ($SO_2$) has been

recognized and studied for decades, recent efforts have explored new pathways as part of an effort to explain rapid formation during severe haze events such as those observed in Beijing. Liu et al. (2020) measured sulfate formation in hygroscopic, pH-buffered aerosol particles and

demonstrated that the oxidation of $SO_2$ by hydrogen peroxide ($H_2O_2$) in aqueous aerosol particles can explain the missing sulfate source during severe haze pollution events. Ge et al. (2021) used

the Community Earth System Model Version 2 (CESM2) to evaluate the effects of in-cloud aqueous-phase reaction mechanisms on $SO_2$ oxidation and the importance for sulfate formation on hazy days.

     Formation of SOA through aqueous-phase chemistry (aqSOA) was argued to potentially be significant by Blando and Turpin (2000) and has since been the subject of numerous

laboratory, field, and modeling studies. Interest in aqSOA is partly due to its potential to help explain discrepancies between observed mass loadings and model estimates that include only gas-phase chemistry (Carlton et al., 2008; Ervens and Volkamer, 2010; Ervens et al., 2011; Guo et al., 2012; McNeill, 2015; Gilardoni et al., 2016). A large fraction of aqSOA is believed to form through photochemistry as water-soluble products of gas-phase chemistry enter cloud

droplets or aerosol liquid water (ALW) and react in the aqueous phase with hydroxyl radical (OH) or other oxidants, with some of the reaction products then remaining in the particle phase after evaporation of the water (Perri et al., 2009; Lim et al., 2010; Liu et al., 2012a; McNeill et al., 2012; Lin et al., 2014). Lamkaddam et al. (2021) found that 50-70 % of the products of gas-phase OH-oxidation of isoprene partitioned into a liquid water film and subsequently reacted

with dissolved OH, resulting in the production of more oxygenated and less volatile products that would remain in the aerosol phase. Aqueous SOA is also produced as aldehydes such as glyoxal and methylglyoxal partition into water droplets and undergo nonoxidative reactions that are not dependent on UV or visible light (De Haan et al., 2009; Galloway et al., 2014), and by the aqueous oxidation of organic compounds by singlet molecular oxygen ($^1O_2^*$), triplet excited

states of organic compounds ($^3C^*$), and hydroperoxyl radicals ($HO_2$) (Smith et al., 2014; Smith et al., 2015; Kaur and Anastasio, 2018). Ye et al. (2020) used results from experiments focused on the aqueous-phase photochemistry of three phenolic compounds to demonstrate the importance of aqueous-phase oxidation of moderately-soluble compounds to SOA formation. Tsui et al. (2017) used an updated version of the Gas-Aerosol Model for Mechanism Analysis

(GAMMA) that includes uptake of isoprene epoxydiols (IEPOX) and subsequent formation of SOA to compare formation of IEPOX SOA in cloud water and aqueous aerosol for simulated laboratory and atmospheric conditions.



Unlike experimental studies of secondary aerosol formation through gas-phase reactions (gasSA), for which realistic atmospheric conditions can more easily be simulated, laboratory

investigation of aqSA mechanisms, products, and yields is usually performed in bulk aqueous solutions with high oxidant and precursor concentrations (Liu et al., 2012b; Lim et al., 2013). The experimental concentrations and conditions often differ from those in the atmosphere, which can introduce uncertainty when results are implemented into multiphase models. Among the sources of constraints on experimental conditions is the lack of suitable and sensitive

measurement and detection technology (Ervens et al., 2011; Spracklen et al., 2011). Some laboratory-based studies of aqueous-phase oxidation have been conducted using cloud chambers, which can offer measurement over a range in temperature and pressure and with artificial solar illumination. Such chambers have been used to study multiphase atmospheric photochemistry with one or more cloud formation and evaporation cycles (Berndt et al., 2007; Wang et al., 2011;

Hoyle et al., 2016). However, challenges of creating an environment in which there is both active photochemistry and a controlled population of cloud droplets has limited the number of such facilities.

Oxidation flow reactors (OFRs) are commonly used to study secondary aerosol formation and evolution, both in the laboratory and in the field (Kang et al., 2007; Lambe et al., 2011;

Keller and Burtscher, 2012; Ortega et al., 2013; Simonen et al., 2017). Photolysis of injected or in situ-formed ozone ($O_3$) inside a typical OFR results in OH concentrations that are orders of magnitude higher than found in the atmosphere. However, almost all OFRs are designed for studying gas-phase chemistry and are not generally suitable for studying aqSA formation because of issues such as temperature gradients caused by the UV lights, wall losses of gases at

high relative humidity (RH), and settling losses of droplets (Li et al., 2015; Huang et al., 2017; Mitroo et al., 2018; Cao et al., 2020). To our knowledge, only one study has been conducted in which a flow-through reactor was used to simulate in-cloud aqSA formation from oxidation of soluble gases produced from gas-phase photochemistry (Lamkaddam et al., 2021). Inside the wet-walled flow reactor used in that study, precursor gases react with OH over a timescale of

minutes as with a standard OFR, whereas the subsequent aqueous-phase oxidation occurs in a thin layer of water surrounding the flow cell over a timescale of several hours for each experiment.



In this work, we describe the Accelerated Production and Processing of Aerosols (APPA) reactor, which is an OFR that can be used to study gas- and aqueous-phase secondary aerosol formation from prescribed concentrations of precursors in the lab and from the complex mixture of gases present in ambient air. Reported here is the design and laboratory characterization of the reactor, including examination of transmission efficiencies and residence time distributions for both particles and gases, size distributions of the droplets used in experiments simulating in-cloud chemistry, UV intensity and spatial variability, and OH exposure ($OH_{exp}$) estimation from measurement of the consumption of $SO_2$ and benzene. The observed growth of seed particles on which droplets formed as sulfate was produced from the $S(IV)-O_3$ reaction is compared with a prediction derived from the cloud model kinetic expression presented in Caffrey et al. (2001). We report the production of SOA from OH-oxidation of benzene with minimal liquid water present (RH = 40 %), with ALW (RH = 85 %), and for simulated in-cloud conditions (RH = 100 %). Example results are also provided from measurements of the composition of secondary aerosol formed from ambient air processed under that same set of humidity and liquid water content conditions. Though not described here, the reactor can also be used to study the impact of cloud cycling on the composition and properties of ambient or generated particles.



## 2 Design and experimental setup

### 2.1 Reactor design and operation overview

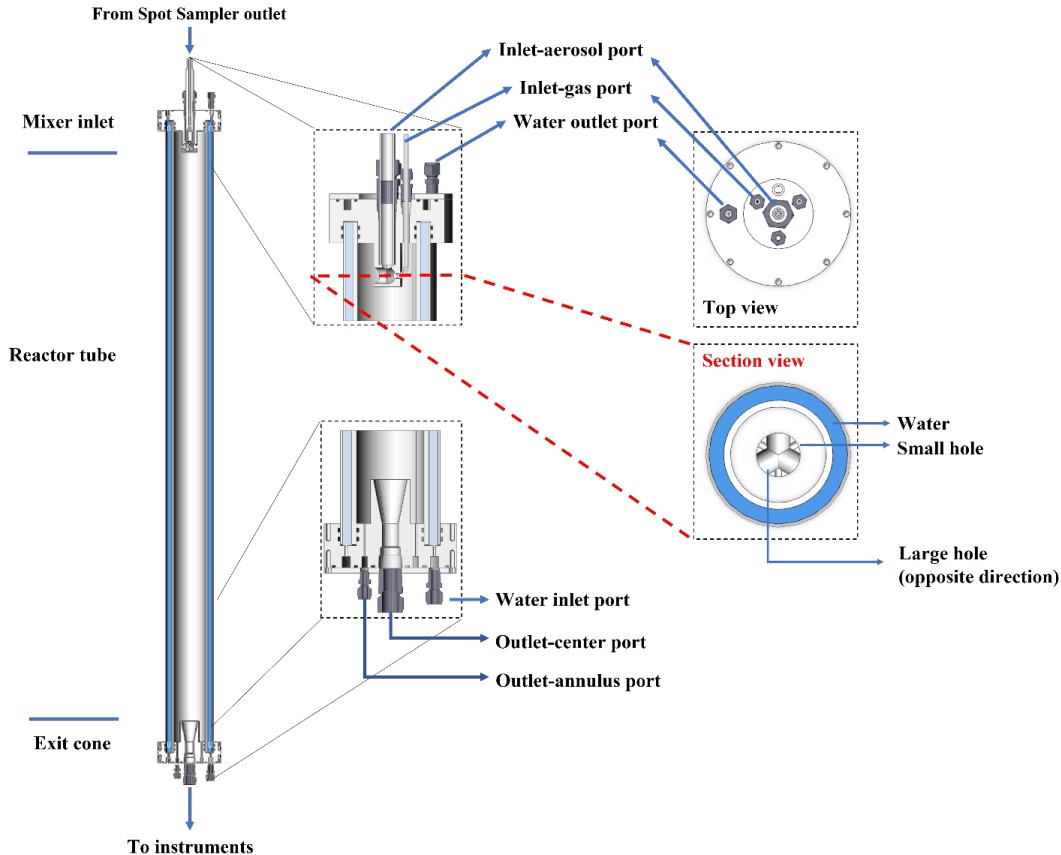

**Figure 1.** Vertical cross-section views of the APPA reactor (left) and horizontal cross-section views of the top cap (right).

A cross-sectional view of the APPA reactor is shown in Fig. 1. The core of the reactor is

a 148 cm L × 8.9 cm OD × 7.8 cm ID PFA Teflon tube (Ametek FPP) with a total internal

volume of 7.5 L, which is identical to that used in the Particle Formation Accelerator OFR

described by Xu and Collins (2021). The PFA tube is surrounded by a 148 cm L × 11.5 cm OD ×

11.0 cm ID cylindrical quartz tube (Technical Glass Products, Inc.), which is used to create a

water jacket as described below. Machined PTFE end caps seal the PFA and quartz tubes at the

top and bottom. The APPA reactor is typically operated as a 254 nm-type OFR, with OH



produced from photolysis of $O_3$ that is produced externally by an $O_3$ generator (Jelight Co., Inc., Model 610) and introduced into the reactor. The reactor is oriented vertically, with inlet flows introduced at the top and outlet flows extracted at the bottom. The nominal total flow rate of 3.0 L min$^{-1}$ results in a mean gas residence time of 150 s. The bottom cap was adapted from the

design used in the reactor described by Xu and Collins (2021). To minimize the influence of the reactor walls and to narrow the particle residence time distribution (RTD), the central 50 % (1.5 L min$^{-1}$) of the total flow is subsampled through the conical sample extraction port and directed to the aerosol and/or droplet analyzers. The outer 50 % (also 1.5 L min$^{-1}$) of the total flow is extracted through eight 0.15 cm holes at the base of the annulus between the reactor ID and the

extraction port in the center of the tube, and is directed to one or more gas analyzers. Those two outlet flows and their corresponding ports in the bottom PTFE cap are hereafter referred to as *outlet-center* and *outlet-annulus*. The flow rates are actively controlled, with dilution or make-up flow used as needed such that they are always 1.5/1.5 L min$^{-1}$ and are unaffected by the flow rates of the sets of analyzers used for different experiments. The reactor system is fully

controlled using National Instruments Labview software and is capable of unattended operation for multiple days.

       To precisely control the reactor cell temperature and to minimize any radial or axial temperature gradients that would promote convective mixing and cause droplet growth or evaporation, temperature-controlled ultrapure water from a chiller (Cole Parmer, Model 10124)

is pumped upward through the 1.0 cm annulus between the outside of the PFA tube and the inside of the quartz tube. To achieve uniform upward velocity around the annulus, the water flows through eight equally-spaced 0.32 cm holes as it enters through the bottom PTFE cap and as it exits through the top PTFE cap. Thermistors measure the temperature of the water entering and exiting the water jacket, with an average difference between the two of only about 0.2 °C

with the solar simulating or UV lamps described below turned on.

       The most novel aspect of the APPA reactor is its use for studying aqueous phase secondary aerosol formation in ALW and in cloud droplets. This is achieved by introducing cloud droplets formed on hygroscopic seed particles at the top and controlling the dew point and temperature in the reactor to produce either a prescribed RH or saturated conditions. To

minimize losses of soluble and reactive gases to liquid water on the walls of the tubing and droplet generator upstream of the reactor, separate flows containing the gas mixture and the seed

aerosol/droplets are used and are rapidly mixed inside the reactor. Those two inlet flows and their corresponding ports in the top PTFE cap are hereafter referred to as *inlet-gas* and *inlet-aerosol*. For the experiments reported here, the inlet-gas flow was controlled to 1.64 L min$^{-1}$ and

the inlet-aerosol flow was 1.36 L min$^{-1}$. The stability of two flows over time is evident in the time series shown in Fig. S1 (a). The inlet-gas flow is subsaturated and particle-free and contains the precursor gas(es) and $O_3$, while the inlet-aerosol flow is typically saturated zero air and, for most experiments, contains droplets formed on monodisperse seed particles. The inlet-aerosol flow is injected through the center injection port shown in the horizontal cross-sectional view of

the top cap in Fig. 1 and the inlet-gas flow is introduced through three equally spaced ports that surround it.

### 2.2. Inlet-aerosol flow and droplet generation

Typical operation of the APPA involves injection of droplets formed on monodisperse seed particles into the top of the reactor and then measurement of the amount, properties, and/or

composition of secondary aerosol that was added to them after they exit from the bottom and are dried. Because an objective of many of the experiments is to contrast aerosol formation in cloud droplets with that when dry or aqueous seed particles are present, droplets are typically injected even when the RH in the reactor is controlled to less than 100 % in order to minimize bias between the different reactor conditions. The flow path and components of the aerosol and

droplet generation system are shown in Fig. 2. To date, most experiments have used potassium sulfate ($K_2SO_4$) seed particles because they are non-acidic, have a dynamic shape factor close to 1, and effloresce at an RH of about 60 % (Freney et al., 2009), which permits measurements without ALW at higher RH than would be possible with common aerosol types such as ammonium sulfate and sodium chloride. The aerosol is generated by atomizing a 0.1 M aqueous

$K_2SO_4$ solution with an atomizer (TSI Inc., Model 3076), drying with a diffusion dryer consisting of a perforated tube surrounded by molecular sieve pellets, and size classifying with a high flow differential mobility analyzer (DMA) (Stolzenburg et al., 1998). The aerosol is charge-neutralized in one soft x-ray neutralizer upstream of the DMA and then again in another downstream of the DMA in order to reduce the charged fraction and resulting electrostatic losses

of the particles. Additionally, static charge on the inside of the reactor is minimized prior to the start of a series of experiments by pushing compressed zero air through a bipolar ionizer



(SIMCO Inc., Model 4012229) and through the reactor. The particle diameter used for most experiments to date is 0.1 μm, which was selected to balance the desire to use smaller particles to maximize the relative change in size accompanying growth from addition of secondary aerosol

and the desire to use larger particles to provide sufficiently high surface area concentration for non-cloud experiments to maximize the fraction of oxidation products that condense on them. For that diameter, the seed particle concentration inside the reactor can be varied between about 3,000 and 30,000 cm$^{-3}$ using an actively controlled dilution flow. For the seed particle concentration of approximately 20,000 cm$^{-3}$ that was used for most experiments reported here,

the resulting surface area concentration was between 1,200 and 1,400 μm$^2$ cm$^{-3}$ for low RH operation with no ALW or cloud droplets.

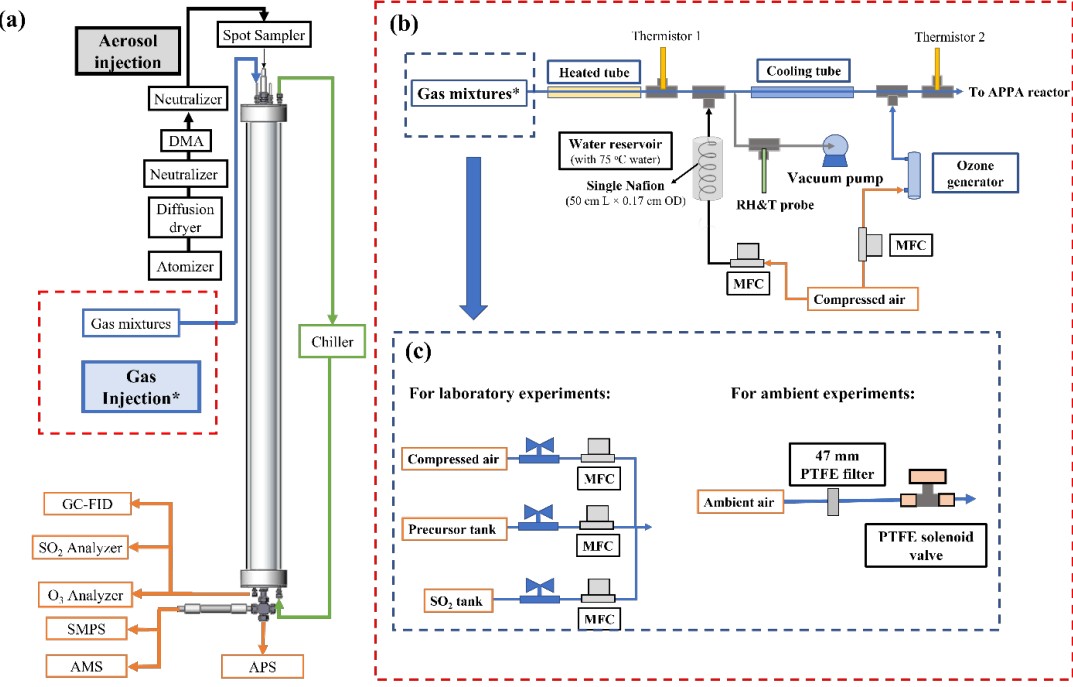

**Figure 2.** (a) Schematic diagram of the APPA reactor and (b) associated experimental setup of the gas mixture injection with (c) configurations shown for laboratory and ambient measurements.




Droplets are formed on the monodisperse particles as they pass through a modified Spot Sampler (Aerosol Devices, Inc., Model 110A) that is positioned on top of the reactor assembly and connected to the top cap through a short interface tube as depicted in Fig. 3. The Spot Sampler uses a three-stage, moderated, laminar flow water condensation growth tube similar to

that described by Hering et al. (2014). Particles activate as they are exposed to a supersaturated environment and grow into droplets with diameters between about 3 and 4 μm, effectively forming a fog (Hering and Stolzenburg, 2005; Eiguren Fernandez et al., 2014). The resulting LWC inside the reactor is between approximately 0.1 and 1.0 g m$^{-3}$ for the range in seed particle concentration described above. The Spot Sampler used for this application was modified by

increasing the bore diameter of the condensation growth tube to about 6 mm and using more powerful fans for the heat sinks on the thermoelectric coolers in the moderator section. The focusing nozzle at the outlet of the standard Spot Sampler was replaced with a machined interface tube that carries the droplets to the inlet-aerosol port. Cooling water is pumped through coiled tubing wrapped around that interface to prevent warming of the flow that would result in

evaporation of the droplets.





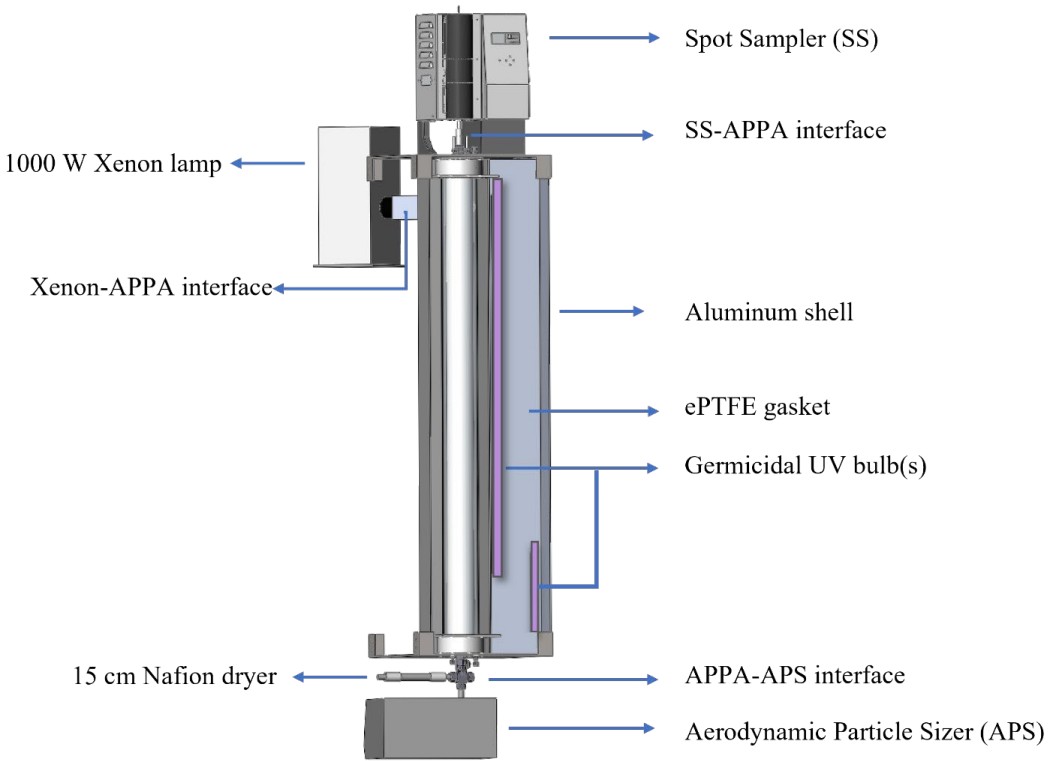

**Figure 3.** Assembly view of APPA reactor.

## 2.3 Inlet-gas flow and RH control

The gas mixture introduced into the reactor contains $O_3$, water vapor, and either

prescribed concentrations of aerosol precursors or particle-filtered ambient air. Almost all tubing

and fittings in the flow path are made of PFA Teflon, with the exceptions being the solenoid

valve, for which all wetted parts are PTFE, two thermistors that are covered in PTFE thread tape,

and, for measurements of ambient air, the 47 mm PTFE membrane filter (Sartorius Stedim

Biotech., Product 36229-44) used to remove ambient particles (the filter housing is PFA). The $O_3$

concentration is controlled using a mass flow controller (MFC; AALBORG Inc., GFC17-

500SCCM) that varies the flow of zero air that is pushed through a generator in which $O_2$ is



photolyzed (Jelight, Inc., Model 610). Because $O_3$ production in the generator is relatively insensitive to the air flow rate through it, a small purge flow is extracted through a critical flow orifice immediately downstream of the generator such that the amount of $O_3$ added to the gas

mixture varies with the difference between the total flow through the generator and that purged through the orifice. Using that approach, the $O_3$ mixing ratio in the reactor can be controlled from 0.1 to 5.0 ppm. For laboratory experiments for which a precursor gas is injected, its concentration is controlled by an MFC (Alicat Scientific, Inc., Model MC-100SCCM) downstream of a pressurized cylinder or tank containing the precursor in a balance of zero air.

The gas mixture is heated (typically to 55 °C) just upstream of the point at which water vapor is added in order to prevent localized saturation and condensation, which could otherwise cause losses of soluble gases. The water vapor concentration is controlled to create saturated conditions or to produce the desired RH in the reactor after mixing with the cool and saturated droplet flow and brought to the controlled reactor temperature. To minimize dilution of the gas mixture flow

for measurements with ambient air, concentrated water vapor is added from a hot, nearly saturated flow that is generated by pushing zero air controlled by an MFC (Alicat Scientific, Inc., Model MC-500SCCM) through a 50 cm L × 0.17 cm OD Nafion tube that is submerged in water inside a stainless tank that is maintained at a fixed temperature (typically 75 °C). Immediately downstream of the tee where the water vapor is added, the mixed flow is forced through a small

orifice to promote efficient mixing. To prevent contact of the gas mixture flow with the RH/T sensor (Vaisala, Model HMP110) that is used to determine the water vapor pressure, a 0.8 L min$^{-1}$ flow is split off and pulled past the sensor and then purged. Just upstream of the reactor, the humid gas mixture is cooled in a segment of the PFA tube that is submerged in a temperature-controlled water bath and is then split between three PFA tubes that extend through the three

inlet-gas injection ports shown in Fig. 1. The three tubes extend into the interior of the reactor where they are press-fit into the outer curved surface of the hollow mixer also shown in that figure. Each of the three parts of the inlet-gas flow is introduced inward and perpendicular to the inlet-aerosol flow entering from above. The three gas mixture flows mix with the droplets and are pushed through holes on the opposite side of the hollow mixer to promote rapid and efficient

mixing, while also minimizing impaction losses and any evaporation/growth of the droplets.





The RH in the reactor cell is calculated from the cell temperature and the water content in the two inlet flows and, independently, from the cell temperature and the water content in the outlet-annulus flow. The calculation based on the inlet flows is used for the RH control, while that based on the outlet flow is used as a check. To date, the cell temperature has typically been

maintained at 20 °C. The flow exiting the Spot Sampler is saturated, with a dew point of approximately the minimum temperature reached by the flow as it exits the base of the cold moderator section. Though the moderator temperature has typically been maintained at the 3 °C lower limit possible, that is the temperature of the growth tube wall, and the air temperature is significantly higher. By mixing the flow from the Spot Sampler with zero air and measuring the

resulting RH in the temperature-controlled reactor cell, it was determined that the flow exiting the Spot Sampler has a dew point of about 14.5 °C. The temperature to which the inlet-gas flow is controlled is optimized for each experiment, but is fixed throughout an experiment to minimize the time needed to switch between measurements at different RH. During a multi-hour to multi-day experiment, the water content in the inlet-aerosol flow and the temperatures of the

inlet-gas and inlet-aerosol flows and of the reactor are all fixed and the cell RH varies only with the water vapor concentration in the inlet-gas flow, which is actively controlled by adjusting the flow rate of zero air pushed through the submerged Nafion tube.

Figure 4 provides a graphical depiction of the inlet-gas flow RH needed to result in the commonly used set of 40 % RH (dry seed), 85 % RH (ALW), and saturated (cloud) conditions

for the typical inlet-aerosol, reactor, and inlet-gas temperatures of 14.5 °C, 20.0 °C, and 31.0 °C, respectively. As shown in the figure, the RH in the inlet-gas flow is controlled to be higher than that needed to result in 100 % RH in the reactor for cloud conditions, which was experimentally found to be necessary to prevent droplet evaporation. Fig. S1 (b) shows an example time series of the dew points of the two inlet flows and the temperature of the reactor for the same set of

conditions depicted in Fig. 4.



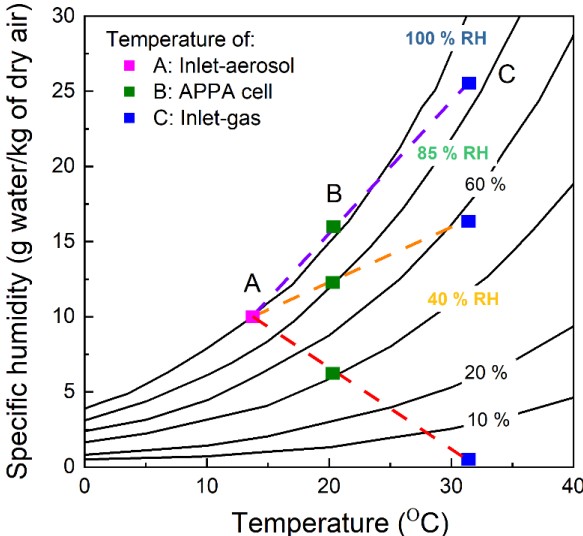

**Figure 4.** Relationship between the temperature and water vapor content of the two inlet flows and the resulting RH in the reactor.

**2.4 Outlet flows and gas and aerosol measurements**

As shown in Figures 2 and 3, a ¾" (1.9 cm) Swagelok cross is mounted directly to the outlet-center port at the base of the reactor. For the orientation shown in the figures, the bottom and right legs of the cross are used only for measuring the size distribution of the droplets with

an aerodynamic particle sizer (APS; TSI, Inc., Model 3321) that is permanently positioned below the reactor. For normal operation, the outlet-center flow containing the processed seed particles or droplets turns 90° in that cross and then immediately enters a 15.2 cm L x 1.7 cm OD Nafion tube (Perma Pure, Model MD 700), where it is dried sufficiently to evaporate the large droplets that would otherwise have high loss rates due to settling and impaction. The flow is then further

dried in a 61 cm L bundle of eighteen 0.17 cm OD Nafion tubes (Perma Pure, Model PD-070-18T) to reduce the RH to below 20 %. The size distribution and non-refractory composition of the initially single-component and monodisperse particles are then measured with a scanning mobility particle sizer (SMPS; fabricated in-house) and an Aerodyne high-resolution time-of-flight aerosol mass spectrometer (HR-ToF-AMS; DeCarlo et al., 2006). When measuring the

droplet size distribution, the 1 L min$^{-1}$ sample flow of the APS is pulled through a thin-walled





stainless steel tube that is press fit on the sample flow inlet of the APS and extends up through the cross and to a point just below the conical extraction section in the reactor cap. For the orientation shown in Figures 2 and 3, 3.5 of the 4 L min⁻¹ humidified sheath flow of the APS is introduced through the right leg of the cross, with the remaining 0.5 L min⁻¹ pulled from the

outlet-center flow around the thin-walled tube carrying the APS sample flow. As with the interface between the Spot Sampler and reactor, that between the reactor and APS is cooled to the temperature of the reactor to prevent droplet evaporation. The outlet-annulus flow is always connected to an $O_3$ analyzer (Teledyne, Model T400U) and always flows past an RH/T sensor (Vaisala, Model HMP110). For some experiments, the flow is also sampled with an $SO_2$ analyzer

(Teledyne, Model T100UP) and/or a gas chromatograph with flame ionization detector (GC-FID; SRI Inc., Model 8610C).

**2.5 Light source and intensity profile**

As depicted in the assembly view in Fig. 3, the reactor is housed in a 158 cm H × 20 cm L × 20 cm W aluminum shell that is in two halves that swing open and shut with hinges. The

light sources are located outside of the quartz tube (which is outside of the PFA reactor). This configuration is similar to that in OFRs with UV lights surrounding a quartz tube reactor such as the Caltech photooxidation flow tube (CPOT; Huang et al., 2017) and Toronto photooxidation tube (TPOT; George et al., 2007; Lambe et al., 2011), with the important difference here that the UV must also be transmitted through the water layer and PFA tube. The quartz tube, ultrapure

water, and PFA tube all absorb little visible or UV radiation (Beder et al., 1971; Litjens et al., 1999; Peng et al., 2017). However, the thick-walled PFA tube is translucent but dull white in appearance and much of the light incident upon it is reflected and not transmitted to reach the interior of the reactor. To maximize the UV intensity and uniformity inside the reactor, all interior surfaces of the aluminum shell are covered by 6 mm thick, highly UV-reflective

expanded PTFE (ePTFE) gasket (Intertech, Inc., Product SQ-S). The intent is to mimic an integrating sphere, with photons repeatedly reflected by the gasket and onto the outer surface of the PFA tube to maximize the fraction that reach the interior of the reactor and to make the UV uniform and nearly isotropic. The spectral intensity inside the reactor is measured using a UV-Vis spectrometer (StellarNet, Model BLK-CXR-SR) through a fiber optic cable that is

terminated with a cosine receptor and permanently secured in a threaded port in the top cap.



For standard operation as an OFR, a pair of 254 nm-emitting germicidal UV bulbs that collectively span the length of the reactor (OSRAM, Model G36T8; 122 cm and USHIO, Model G10T8; 46 cm) are mounted on the inside of the ePTFE gasket as shown in Fig. 3. The output of the UV lamps is computer controlled using a dimmable lamp ballast and is typically maintained

at a level for each RH that results in loss of $15\pm2$ % of the added $O_3$ between the top and bottom of the reactor, which represents a balance between maximizing the $OH:O_3$ concentration ratio and minimizing the $O_3$ (and consequently OH) gradient over the length of the reactor.

An alternate use of the APPA reactor is for studying the impact of solar radiation on biological particles or other particle types. To achieve this, a 1000 W xenon lamp (Newport Inc.,

Model 66924-1000XF-RI) is mounted externally, with its focusing lens housing inserted into one end of a 7.6 cm × 7.6 cm square tube interface that leads to a to a hole through the aluminum shell and ePTFE gasket surrounding the reactor. That interface is attached to the aluminum shell at a 45 degree angle so that the focused beam is incident upon the inner surface of the ePTFE gasket and not the quartz tube in order to minimize local heating and to maximize light intensity

uniformity. A 7.6 cm × 7.6 cm B270 or borosilicate glass window is secured in the interface to produce a spectrum that most closely matches the solar spectrum, especially in the short UV wavelength range close to 300 nm. The light intensity inside the reactor can be adjusted over a wide range through a combination of adjustment of the voltage output of the xenon lamp controller, partial blocking of the beam using a sliding baffle in the interface tube, and swapping

the standard 1000 W bulb with an interchangeable 450 W bulb.





## 3 Result and discussion

### 3.1 Reactor characterization

3.1.1 Gas and particle transmission efficiencies


The particle transmission efficiency through the reactor was evaluated with continuous injections of size-classified ammonium sulfate (AS) particles with mobility diameters ranging from 0.050 to 0.20 μm. The transmission efficiency was calculated as the ratio of particle concentrations downstream of the reactor and downstream of a 150 cm L × 0.95 cm OD copper

tube bypass measured with a condensation particle counter (CPC; TSI Inc., Model 3760A). As with the gas transmission efficiency tests described below, the flow rate through the reactor was the same 3.0 L min⁻¹ used during normal operation. The measurements were repeated 2 or 3 times for each particle size, with agreement between measurements found to be to within ±5 % for each diameter. The particle transmission efficiency increases with increasing particle size,

from 0.67 for 0.050 μm particles, to 0.94 at 0.080 μm, and 0.98 at 0.20 μm. As shown in Fig. 5 (a), the size-dependent particle transmission efficiency through the APPA reactor is similar to that for the OFR described by Xu and Collins (2021), which is not surprising given the similarity in the materials and designs of the reactor tubes and end caps of the two. Figure 5 (b) compares the particle transmission efficiencies of the APPA OFR and several flow tube reactors with non-

metal wall materials. The potential aerosol mass (PAM) reactor for which data are provided is the 15 L glass cylindrical chamber described in Lambe et al. (2011). The particle transmission efficiency of the APPA reactor is significantly higher than those of the quartz PAM, TPOT, and CPOT, though somewhat lower than those of the Environment and Climate Change Canada OFR (ECCC-OFR; Li et al., 2019) and the TUT Secondary Aerosol Reactor (TSAR; Simonen et al.

(2017) at the smallest particle sizes evaluated. The exact causes of the differences in transmission efficiencies among the reactors are unknown, though subsampling of the center flow at the outlet of the APPA likely contributes to its higher efficiency relative to the quartz PAM, TPOT, and CPOT, while its lower efficiency relative to the ECCC-OFR and TSAR may in part be due to differing residence times (150, 120, and 40 s for the APPA, ECCC-OFR, and

TSAR, respectively).

Gas transmission efficiency was evaluated for SO₂, O₃, and CO₂, which were selected as representative of gases that adhere to, react on, and are unaffected by reactor walls, respectively



(Lambe et al., 2011; Ahlberg et al., 2017; Huang et al., 2017). Transmission efficiencies were calculated as the ratios of the $SO_2$, $O_3$, and $CO_2$ concentrations measured downstream and

upstream of the reactor using the $SO_2$ and $O_3$ analyzers described in Section 2.4 and an NDIR analyzer (Li-COR Biosciences, Model Li-840A) for $CO_2$. Prior to measurement of the $SO_2$ transmission efficiency, the $SO_2$ gas mixture was pushed through the reactor for about 20 min to passivate the tubing and reactor surfaces, following the approach described by Lambe et al. (2011). Figure 6 shows the gas transmission efficiencies for 0 % < RH ≤ 100 %. The

transmission efficiencies of $CO_2$ and $O_3$ were over 90 % over the RH range tested for each. For $SO_2$, transmission decreases from over 90 % at an RH of 40 % to 0.8 and 0.73 at RH of 85 % and 100 %, respectively. For comparison, Lambe et al. (2011) reported that the measured $CO_2$ and $SO_2$ transmission efficiencies for the TPOT were $0.97 \pm 0.10$ and $0.45 \pm 0.13$, respectively, and for the quartz PAM were $0.91 \pm 0.09$ and $1.20 \pm 0.40$, respectively.


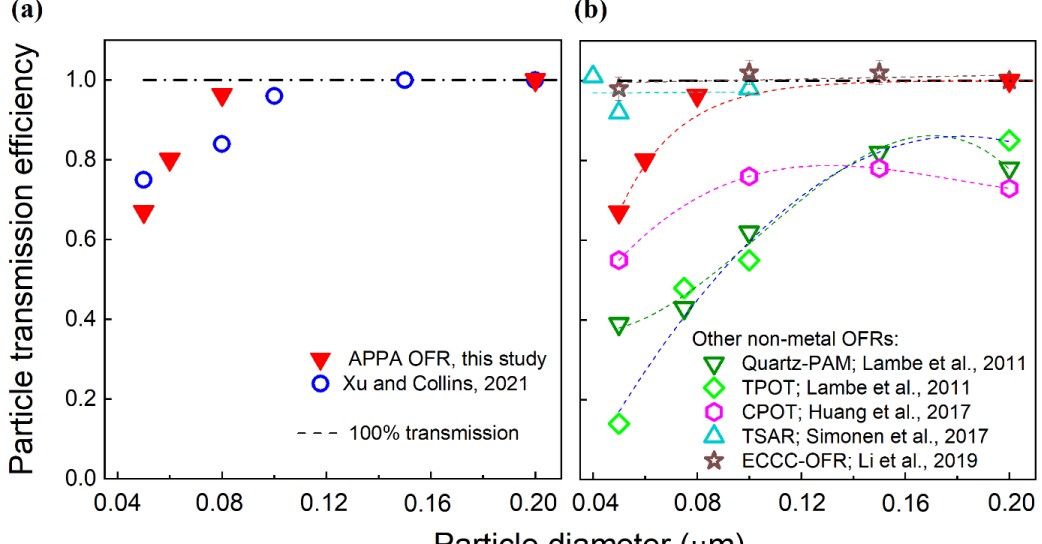

**Figure 5.** Particle transmission efficiency of the APPA reactor compared with those reported **(a)** for the OFR described by Xu and Collins (2021) and **(b)** for several non-metal OFRs reported in the literature, as described in the text.





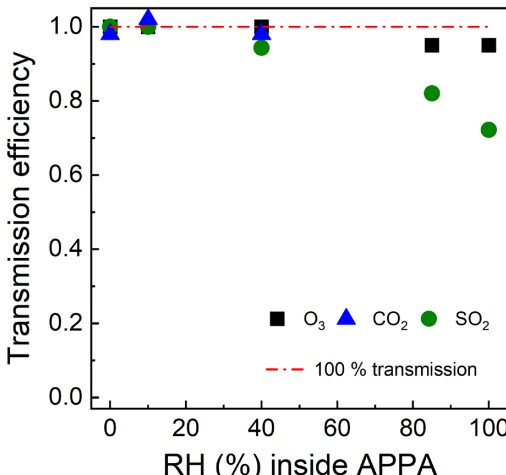

**Figure 6.** Gas transmission efficiencies of the APPA reactor as a function of relative humidity.


### 3.1.2 Gas and particle residence time distributions

Though the extent of processing of gases and particles inside an OFR is typically reported as a single value such as OH exposure or equivalent photochemical age, developed flow velocity profiles and mixing due to convection and/or flow cell geometry lead to a continuum of residence times and corresponding extent of processing. The spread in exposure is typically reported as a residence time probability distribution function, as described in by (Mitroo et al.,

2018). Such functions are often referred to simply as residence time distributions or RTDs.

The approach used to characterize the RTDs of particles and gases is the same as that described in Xu and Collins (2021). Briefly, an MFC was used to introduce 10 s pulses of either 0.20 µm AS particles or pure $CO_2$. The particle and $CO_2$ concentrations in the outlet-center flow were measured with the CPC and $CO_2$ analyzers identified in the previous section. Both the

particle and gas RTD measurements were repeated three times. The resulting distribution functions for particles and gas are shown for the APPA and other reactors in Figures 7 (a) and 7 (b), respectively. The combination of a relatively small inner diameter, a uniform and constant reactor temperature maintained by the water jacket, and the subsampling of the center flow at the outlet results in gas and, especially, particle RTDs of the APPA that are narrow relative to the

other reactors included in the figures and that lack the long tail expected even for ideal laminar flow.

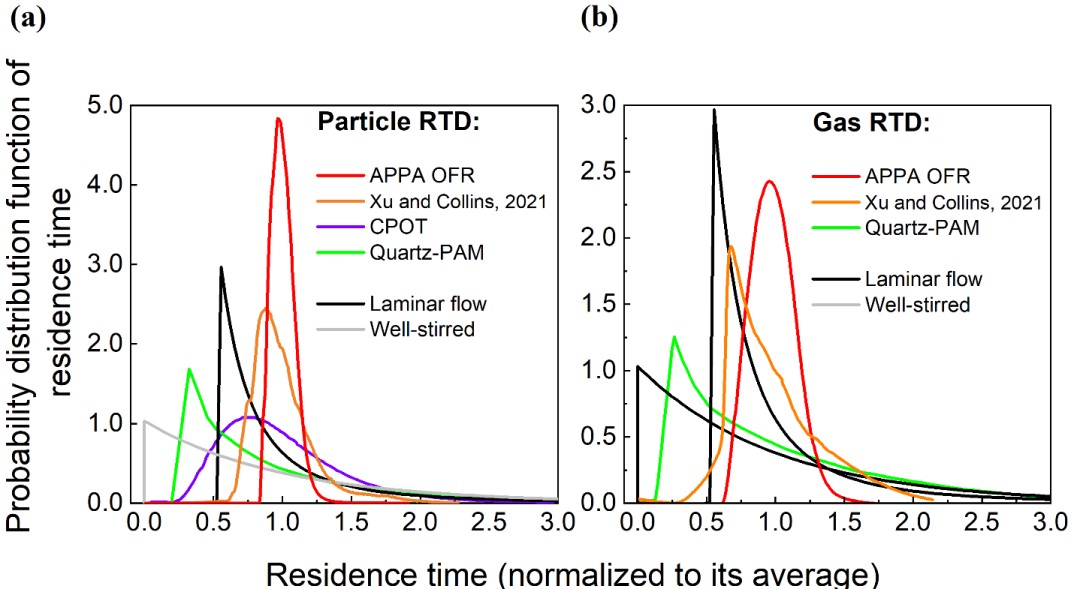

**Figure 7.** Residence time probability distribution function of **(a)** particles and **(b)** gases in the APPA and in other
reactors as reported for the PAM by Lambe et al. (2011), CPOT by Huang et al. (2017), and PFA by Xu and Collins
(2021) and downloaded from PAMWiki (2022).

### 3.1.3 Droplet size distribution and temperature control

Droplet size distributions measured at the outlet of the APPA by the APS are shown in
Figure 8. As is true for most of the experiments reported here, the droplets formed inside the
Spot Sampler on 0.1 µm diameter $K_2SO_4$ particles and were introduced into the top of the reactor
in the 1.36 L min$^{-1}$ inlet-aerosol flow. For these characterization experiments, the seed particle,
and consequently droplet, concentration was varied from 20,000 to 70,000 cm$^{-3}$. The measured
droplet size distributions shown in Fig. 8 are normalized by the integrated concentrations to
emphasize changes, or lack thereof, in shape and peak location with varying concentration. The
mean diameter of the droplets is stable at approximately 3.3 µm for the range in concentration
examined here. As shown in Fig. S3, the shape of the droplet size distribution was also stable
over a period of several months, with the mean diameter varying by only ± 5 %. For the 20,000
cm$^{-3}$ concentration used for most experiments, the resulting LWC is approximately 0.3 g m$^{-3}$.
Compared with the droplet size distribution measured directly from the Spot Sampler, which is



similar to that reported by (Lewis and Hering, 2013), the distribution measured at the outlet of the APPA has a tail at the left side, which is thought to be caused by partial evaporation of droplets near the walls in the interface between the Spot Sampler and inlet-aerosol port. The

efficiency with which the droplets were transmitted through the reactor and the Nafion tube and bundle was found to be over 80 % from experiments in which the seed particle concentration upstream of the Spot Sampler was measured with a CPC (Aerosol Device Inc., Model MAGIC 210) and compared with that calculated from the size distribution measured by the SMPS located downstream of the APPA and Nafion dryers.

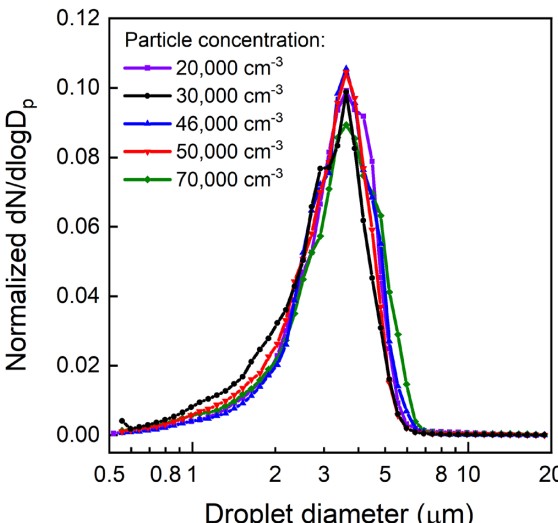

**Figure 8.** Droplet size distributions measured at the outlet of the APPA reactor for a range in concentration.


### 3.1.4 Sulfate formation in cloud droplets

Though, like most OFRs, the APPA reactor is most often used to measure aerosol formation resulting from OH chemistry, conditions inside the reactor during cloud chemistry experiments were first evaluated through the well-studied oxidation of dissolved $SO_2$ by $O_3$, leading to formation of sulfuric acid and growth of the particles on which the droplets formed. The experimental conditions for these tests differed from those for standard operation only in

that the UV lights were not turned on and the diameter of the $K_2SO_4$ seed particles was varied.





The use of ~pH-neutral $K_2SO_4$ for these experiments minimized the influence of the seed particles for the highly pH-dependent reaction. The $SO_2$ and $O_3$ mixing ratios at the top (inlet) of the reactor were fixed at 50 ppb and 1.5 ppm, respectively.

Figure S2 shows the initial and cloud-processed dry particle size distributions measured with the SMPS when 0.040 μm diameter seed particles were injected. Figure 9 summarizes the relationship between the mode diameters of the initial and cloud-processed particles for that experiment and for others that differed only in the size of the injected $K_2SO_4$ seed particles, with 0.030, 0.050, and 0.10 μm particles observed to grow to 0.0418, 0.0569, and 0.102 μm, respectively. Also shown in that figure are estimates of the particle growth from a 0-D model

that includes reactions for this system as described by Caffrey et al. (2001) and that assumes a cloud droplet diameter of 3.3 μm. The measured and modeled dry diameters of the cloud-processed particles agree within 5 %.

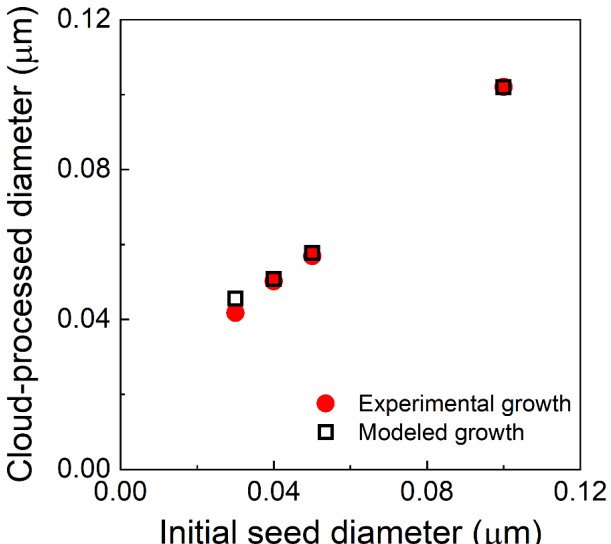


**Figure 9.** Measured diameters of cloud-processed and evaporated particles as a function of the diameter of the injected $K_2SO_4$ seed particles (red solid points), and the expected diameters calculated for the reactor conditions (3.3 μm droplet diameter, 50 ppb $SO_2$, and 1.5 ppm $O_3$) and the set of relevant aqueous-phase reactions (black hollow squares).


### 3.1.5 Light intensity spectrum and profile

As described in Section 2.5, a 1000 W xenon lamp is used instead of the 254-nm mercury lamps for experiments such as those designed to assess the germicidal efficacy of solar radiation.
Spectral irradiances shown in Figure 10 were measured i) inside the reactor and ii) outdoors with the fiber optic cosine receptor pointed at the sun around noon on a sunny day in April. The data are normalized to more clearly show the similarity in spectral shapes. Actinic fluxes were calculated from the measured irradiance spectra and approximate actinic flux to irradiance ratios for nearly isotropic (reactor) and mostly direct (solar) radiation taken from Hofzumahaus et al.
(1999). Actinic flux is the radiant quantity used to calculate photodissociation rates that are used to describe the photochemistry of the atmosphere and is also the most relevant quantity for many biological systems (Kylling et al., 2003). The ratio of the reactor and outdoor actinic fluxes is also included in Figure 10, with an average inside the reactor of 1.9 times that in direct sun for 300 nm $\leq \lambda \leq$ 400 nm.

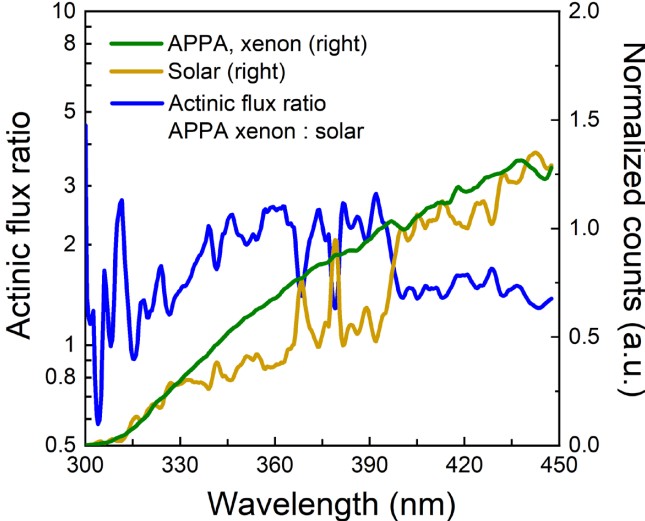


**Figure 10.** Spectral actinic fluxes inside the reactor when illuminated by the xenon lamp (green curve) and outdoors around noon on a sunny day in April (yellow curve), and the wavelength-dependent ratio of the two (blue curve). The approach used to calculate actinic flux from spectral irradiance measured with a spectrometer is described in the text.


The uniformity of the 254 nm UV from the germicidal mercury lamps was evaluated by attaching the receptor of the fiber optic-coupled spectrometer to a metal rod that was inserted through the outlet-center port and moved to five approximately evenly spaced positions between the top and bottom of the reactor. The UV photon counts and normalized intensity at 254 nm as a

function of position are shown in Fig. 11 (a). The 254 nm intensity varies by approximately $\pm$ 10 % throughout the reactor.

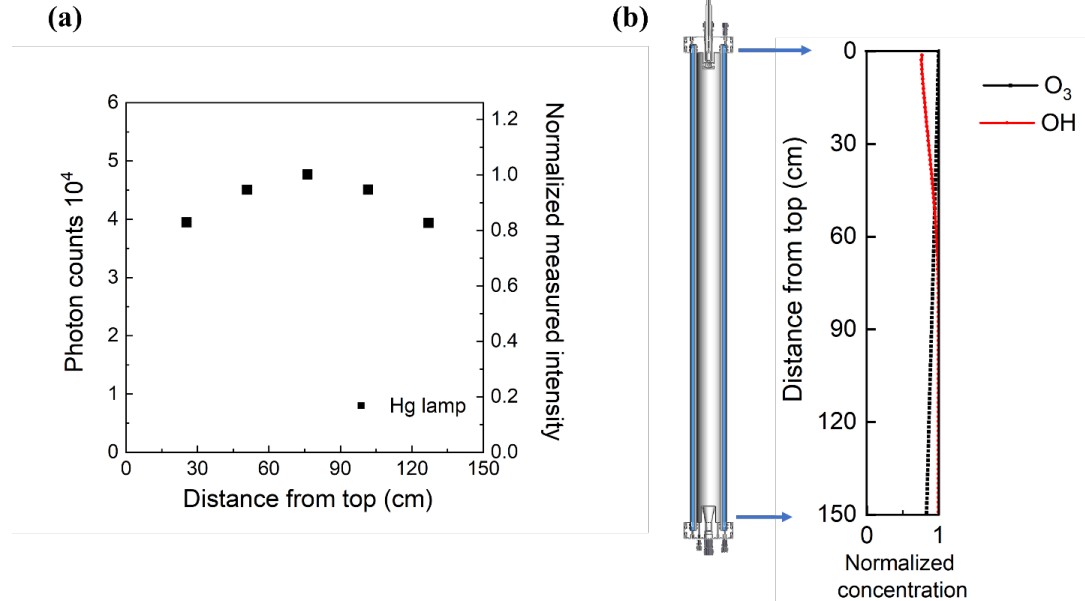

**Figure 11. (a)** Relative UV intensity profile and **(b)** normalized ratio of $O_3$ and OH concentration as a function of

position in the reactor cell.

The rate of OH formation from $O_3$ photolysis at any position in the reactor is dependent upon the local 254 nm UV intensity and the local water vapor and $O_3$ concentrations. The water vapor concentration varies minimally inside the reactor while, as noted above, the UV lamp

output is generally set at a level that results in loss of ~15 % of the added $O_3$ between the inlet and outlet. Though the OH formation rate can be estimated from the UV and $O_3$ profiles, estimating the more relevant OH concentration profile requires consideration of reactions involving an array of radicals and other species. Here, the position-dependent concentrations of OH and other important species were simulated using the KinSim chemical-kinetics solver (Peng





et al., 2016; Peng and Jimenez, 2019). Environmental parameters such as temperature and RH
        and a scaling array for the 254 nm UV intensity based on the data shown in Figure 11 (a) were
        provided as inputs. The reactions used in the simulator included the default set of gas-phase
        reactions in the KinSim "OFR radical chemistry" module, relevant photolysis rate constants for
        254 nm UV, and the aqueous-phase reactions listed in Table S1, for which effective reaction rate

constants were calculated for an LWC of 0.3 g m$^{-3}$ and assuming aqueous phase concentrations
        are described by Henry's Law. The peak 254 nm photon flux specified in the model for each RH
        was then iteratively determined such that the simulated loss of $O_3$ matched that measured. The
        photon fluxes determined in that way ranged from $\sim 4.6 \times 10^{14}$ photons cm$^{-2}$ s$^{-1}$ for 100 % RH
        measurements (2.3 V sent to the adjustable lamp ballast) to $1.1 \times 10^{15}$ photons cm$^{-2}$ s$^{-1}$ for 40 %

RH measurements (3.0 V sent to ballast). The UV intensity required to result in the same
        fractional loss of $O_3$ is higher at low RH because a larger percentage of the $O(^1D)$ produced from
        $O_3$ photolysis undergoes collisional deactivation to form $O(^3P)$, which subsequently reacts with
        $O_2$ to reform $O_3$. Figure 11 (b) shows an example of the profiles of simulated concentrations of
        $O_3$ and OH through the length of the reactor, each normalized by its maximum concentration.

The OH concentration increases with time (and distance from the inlet) over roughly the top 1/3$^{rd}$
        of the reactor and is nearly constant through the lower 2/3$^{rd}$.

        The oxidizing environment inside OFRs is often expressed as the OH exposure ($OH_{exp}$),
        which is normally defined as the product of the average OH concentration in cm$^{-3}$ and the mean
        residence time of the sample flow in seconds. Here, $OH_{exp}$ was calculated from the reactive loss

of $SO_2$ and benzene gas as a function of UV intensity, RH, and added $O_3$ concentration. Those
        experimentally determined values were compared with estimates from the KinSim model with
        the photon fluxes specified above. Reactive loss of $SO_2$ was used to determine $OH_{exp}$ at 40 % RH
        and that of benzene to determine $OH_{exp}$ at 40 %, 85 %, and 100 % RH. Sulfur dioxide was not
        used for the high RH measurements because oxidation by $O_3$ and OH in the ALW or cloud

droplets would cause a high $OH_{exp}$ bias. Sulfur dioxide and benzene were used because their OH
        reaction rate constants of $1.3 \times 10^{-12}$ and $1.23 \times 10^{-12}$ cm$^3$ molec$^{-1}$ s$^{-1}$, result in significant, but not
        complete, reactive loss over the $OH_{exp}$ range of interest. For both $SO_2$ and benzene, mixing ratios
        of between 150 ppb and 250 ppb were injected and the those of the air exiting the chamber were
        measured by the $SO_2$ analyzer and GC-FID, respectively. The reactive loss was determined from

the ratio of the concentration measured with the UV lights on to that with them off. As with





normal operation of the reactor, $OH_{exp}$ was varied by changing the added $O_3$ concentration, with the UV lamp intensity at each RH fixed (and, again, selected to result in the loss of approximately 15 % of the $O_3$).

The points in Figure 12 represent $OH_{exp}$ calculated from measured $SO_2$ loss as a function of initial $O_3$ concentration for a series of experiments at 40 % RH. The two dashed curves in that figure are $OH_{exp}$ estimated from KinSim for the experimental conditions. For the simulations resulting in the values along the upper (black) curve, the only source of "external" OH reactivity ($OH_{ext}$) (Peng et al., 2015) was the added $SO_2$. For the values shown with the lower (red) dashed curve, an additional reactant and reaction were included in KinSim that added 2.0 s$^{-1}$ external

OH reactivity without directly affecting any other species. Though it is unknown whether that change improves agreement because of the presence of one or more species that react with OH, because of loss of OH to the walls, and/or because of sources of error in the experiments and simulations, it is assumed to improve simulation for all conditions and is included in KinSim for all other experiments reported here as well.

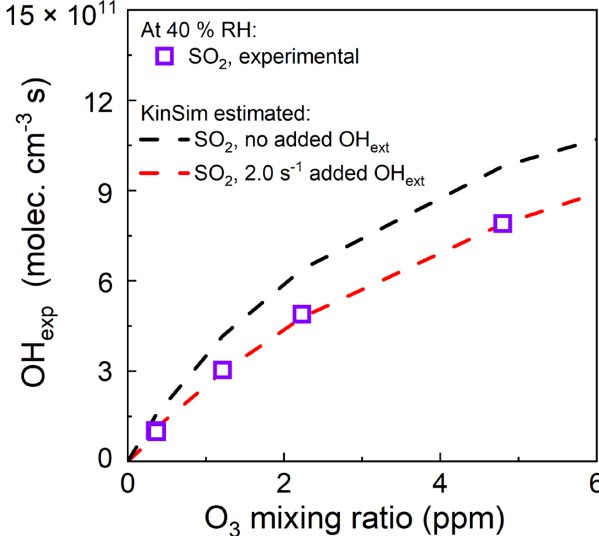


**Figure 12.** Variation of $OH_{exp}$ estimated from $SO_2$ decay as a function of initial $O_3$ mixing ratio for 40 % RH.



The OH$_{exp}$ values calculated from measurement of the reactive loss of benzene for RH = 40 % RH (dry seed particles), 85 % RH (aqueous seed particles), and 100 % RH (cloud droplets) are presented as the markers in Figures 13 (a), (b), and (c), respectively. Whereas the sulfuric acid resulting from the oxidation of SO$_2$ by one OH radical does not undergo any subsequent reaction with OH, oxidation of most organics results in a multi-generation cascade of products that are also reactive with OH. The distribution of products and their OH reaction rate constants are generally unknown. The upper curves in Figure 13 represent OH$_{exp}$ from KinSim simulations

in which OH reacts only with the added benzene, while the lower curves are from simulations in which OH also reacts with the first-generation oxidation products with a reaction rate constant of 10x that of OH reaction with benzene. Reaction of 1$^{st}$ and higher generation oxidation products is expected to increase with increasing OH concentration, resulting in the sort of downward shift in experimentally determined OH$_{exp}$ towards and beyond the lower curve with increasing O$_3$

concentration (and OH production). Additional reaction of oxidation products in the aqueous phase may explain the slightly greater downward shift in data in Figure 13 (c) for the 100 % RH experiments. Based on the comparisons of the experimental and simulation OH$_{exp}$ for all of the SO$_2$ and benzene experiments, a reasonable estimate of uncertainty in the OH$_{exp}$ estimated from KinSim is approximately ±20 %.


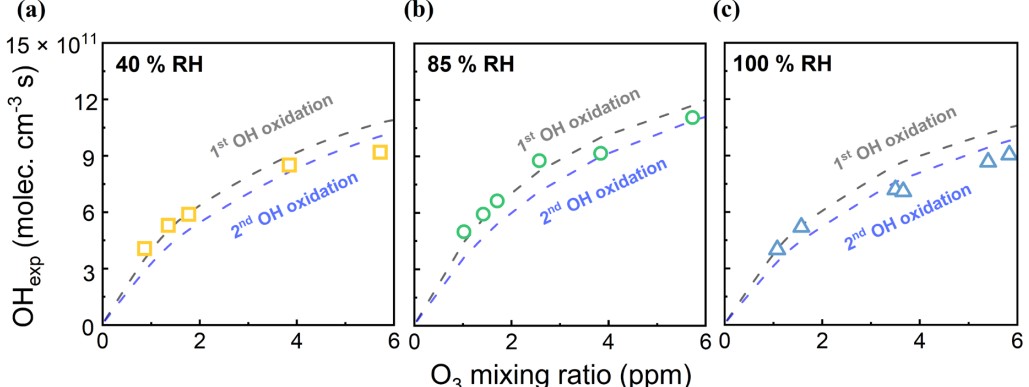

**Figure 13.** Variation of OH$_{exp}$ as a function of initial O$_3$ mixing ratio for **(a)** low RH mode (40 %) **(b)** high RH mode (85 %) and **(c)** cloud mode (100 %).






### 3.2 Measurement of secondary aerosol formation

3.2.1 Gas- and aqueous-phase SOA formation from oxidation of benzene

Secondary organic aerosol formation from a single precursor was studied by injecting between 94 and 101 ppb of benzene and then measuring the growth of the added $K_2SO_4$ seed particles over a range in both $OH_{exp}$ and RH. An example of a set of number size distributions of seed particles without and with added SOA are shown in Fig. 14. The nucleation mode evident in the distributions measured when dry seed particles were present in the reactor (40 % RH) and when ALW was present (85 % RH) were often observed, but typically contributed negligibly to

the total mass concentration.

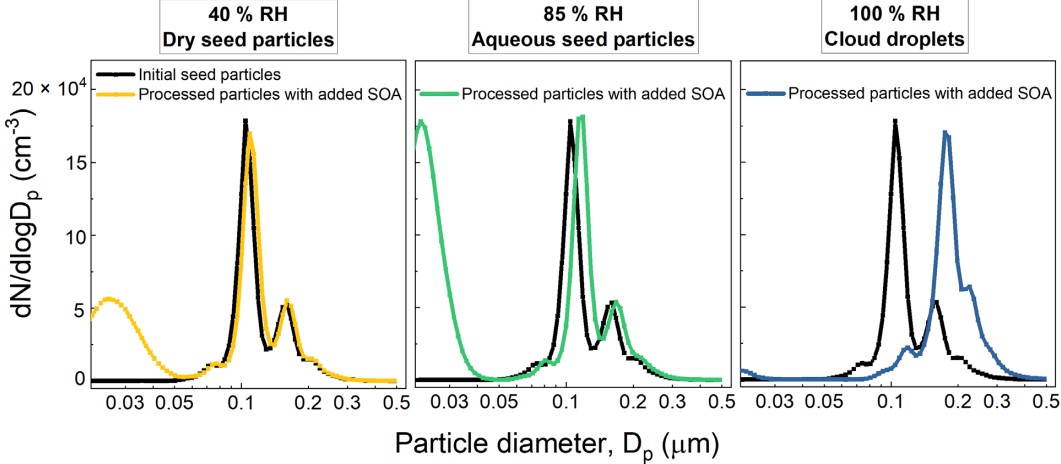

**Figure 14.** Example of a set of number size distributions of seed particles without and with added SOA formed from OH-oxidation of benzene for 40 % RH (left), 85 % RH (middle), and 100 % RH (right). The initial benzene mixing ratio was 90 ppb and the $OH_{exp}$ estimated from KinSim is $3 \times 10^{11}$ molec. cm$^{-3}$ s.

The SOA mass concentration was calculated from distributions such as those in Fig. 14 from the increase in aerosol volume concentration above that of the seed particles and an assumed SOA density of 1.3 g cm$^{-3}$ (Schnitzler et al., 2014). SOA yields were then calculated as the ratio of the mass concentration of SOA to the mass concentration of reacted benzene, which was measured with the GC-FID. Figure 15 summarizes the SOA yields as a function of RH and

$OH_{exp}$. As shown in that figure, the SOA yield for each RH increased with $OH_{exp}$ up to the maximum of $1.04 - 1.20 \times 10^{12}$ molec. cm$^{-3}$ s, which corresponds to a photochemical age of 8 –





9 days for an assumed average atmospheric [OH] of $1.5 \times 10^6$ cm$^{-3}$. Also evident in the figure is

the significant dependence of the yield on the presence and amount of liquid water. Maximum

yields of 0.18, 0.43, and 0.59 were measured when dry seed particles, aqueous seed particles, and

650        cloud droplets were present in the reactor, respectively. Future studies will evaluate SOA

formation from other precursors and will add more comprehensive measurements of the

chemical composition of the aerosol- and gas-phase species formed in the presence of varying

amounts of liquid water.

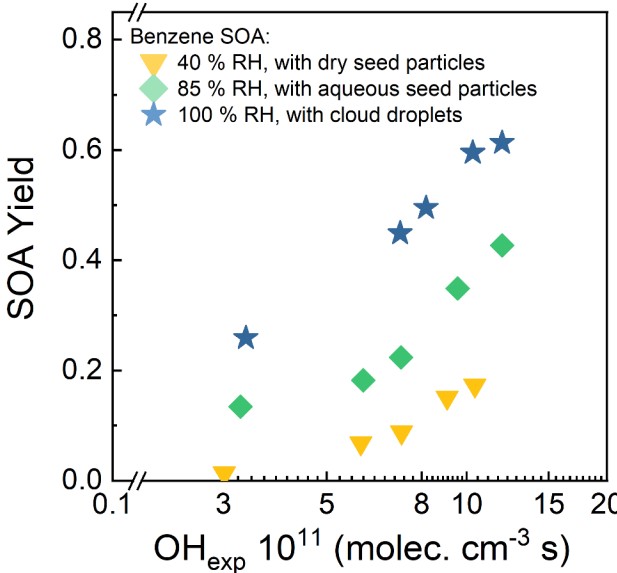


**Figure 15.** Benzene SOA yields as a function of OH$_{exp}$ in low RH (40 %), high RH (85 %), and cloud (100 %)

modes.

3.2.2 Aerosol formed from oxidation of ambient air


The potential contribution of aqueous-phase chemistry to secondary aerosol formation

potential of ambient air is assessed by cycling through RH (and liquid water content) as quickly

as possible to minimize the influence of changing ambient concentrations between

measurements. Approximately 15 min is required for measurement at each RH/OH$_{exp}$, which

665        includes time to reach steady-state and then time to measure two size distributions with the

SMPS, with aerosol composition often simultaneously measured with the AMS. Measurement at

several OH$_{exp}$ as with the benzene experiments summarized in Fig. 15 would take too long for



most study locations. The example RH time series shown in Fig. 16 is a segment of a period of

several weeks during which ambient air was processed through the APPA at the standard RH

steps of 40 %, 85 %, and 100 % and with OH$_{exp}$ steps corresponding to approximately 8 hours

and 4 days photochemical age. That repeated matrix of 6 RH/OH$_{exp}$ pairs required approximately

1.75 h, which includes a few minutes of flushing with dry air after the 100 % RH measurements

to ensure no liquid water remains on the reactor walls.

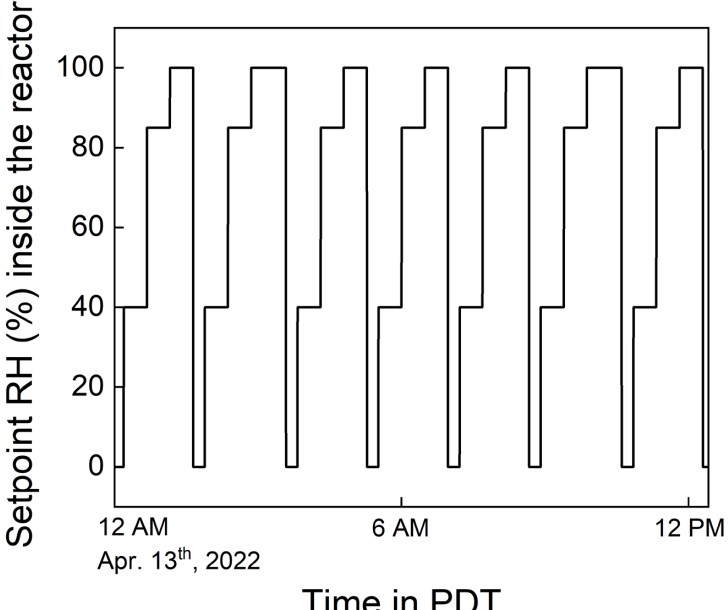

**Figure 16.** A typical RH cycle showing steps in the sequence of 40 %, 85 %, and 100 % RH. The short periods with

very low RH follow the 100 % RH measurements and are designed to evaporate any residual liquid water from the

walls before the start of the next 40 % RH measurement.

Figure 17 provides an example of the influence of aerosol liquid water and cloud water

on the composition of the organic fraction of secondary aerosol that formed as ambient air in

Riverside, CA was exposed to a fixed OH$_{exp}$ of approximately $5 \times 10^{11}$ molec. cm$^{-3}$ s (~3.8 days

photochemical age). As shown in the figure, the O:C ratio calculated from the high resolution

AMS data increased significantly with increased liquid water content, from an average of 0.34

when only dry seed particles were present, to an average of 0.64 with aqueous seed particles, and

an average of 0.89 with cloud droplets. This sort of enhancement in O:C in aqSOA is among the





possible explanations for the frequent observation that ambient aerosol has a higher O:C than that formed in environmental chambers (Reinhardt et al., 2007; Chhabra et al., 2011). Results from continuous processing of ambient air over periods of weeks will be presented in future publications.

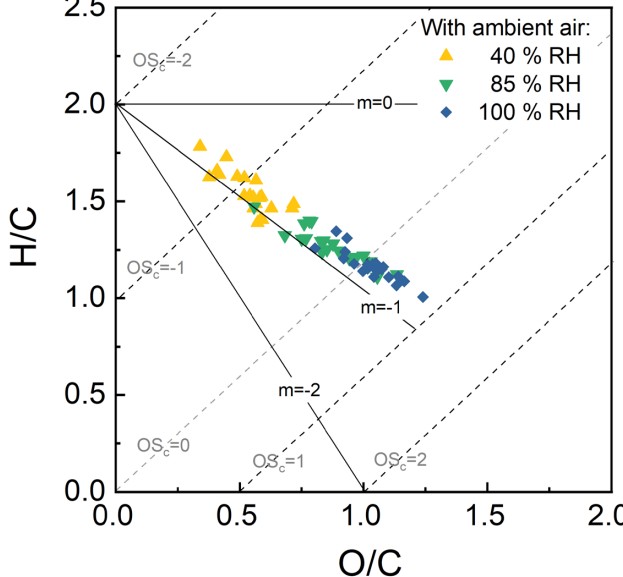


**Figure 17.** O/C and H/C ratios determined from AMS measurements of SOA formed as ambient air was processed in the reactor. The significant decrease in H/C ratio and increase in O/C ratio with the progression from no liquid water in the reactor (40 %) to ALW (85 %) to cloud droplets (100 %) suggests oxidation in the aqueous phase was important.






## 4 Summary

A new all-Teflon flow cell reactor was developed to study i) secondary aerosol formation from gas- and aqueous-phase chemistry and ii) changes in aerosols resulting from cloud processing or exposure to simulated solar or other light sources. To date, the Accelerated Production and Processing of Aerosols (APPA) reactor has primarily been used as an oxidation flow reactor, with photolysis of externally generated $O_3$ providing an OH exposure of between $8 \times 10^{10}$ molec. $cm^{-3}$ s and $1.2 \times 10^{12}$ molec. $cm^{-3}$ s over the ~150 s mean gas residence time. The

geometry, inlet and outlet designs, and tight temperature control result in minimal mixing and a narrow residence time distribution. The most unique aspect of the reactor is the ability to vary the liquid water content present in aqueous aerosol or ~3.3 μm diameter cloud droplets that are formed on monodisperse seed particles and flow through the reactor together with the $O_3$, OH, and reactive precursor gases. A set of measurements for a prescribed gas mixture or ambient air

can thus investigate the amount, properties, and composition of secondary aerosol formed across a matrix of conditions spanning both $OH_{exp}$ and RH/LWC. The experimental system is fully automated and designed for continuous operation over extended periods of time. A series of experiments and numerical simulations summarized here explored the characteristics and capabilities of the reactor system. Example results reported here provide a preview into ongoing

work investigating the roles of aerosol liquid water and cloud water in aerosol formation for i) a range of organic precursor gases and ii) ambient air over multi-week sampling periods.

### Data availability
Data presented in this work are available from the authors.

### Supplement
The supplement related to this article is available.

### Author contributions
DRC designed the reactor and edited the paper. NX performed the experiments and simulations, processed the data, and wrote the paper.

### Competing interests
The authors declare that they have no competing interests.



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
