# Peer review of "An oxidation flow reactor for simulating and accelerating secondary aerosol formation in aerosol liquid water and cloud droplets"

_Atmospheric Measurement Techniques, 2022_

## Author Comment (AC1)

The authors present a new flow reactor designed for investigating multiphase chemistry in aerosols and cloud droplets. This is an interesting and technically challenging idea, which could potentially be an important tool to gain insights into these understudied processes. Care was taken in the design of the reactor to avoid temperature gradients due to the UV lamps, which would interfere with multiphase partitioning, to minimize losses of soluble gases, and to strictly control RH. They use a Spot sampler to generate cloud-like droplets from seed aerosol for study in the reactor. They present preliminary results for studies of the influence of aqueous processing on SOA formation in the lab and using ambient air. I believe the manuscript is publishable in AMT after some minor issues are addressed and the manuscript is revised accordingly.

- The article is too long, with 17 figures in the main text. Surely this can be tightened up, with some material moved to SI. At the same time the most interesting figure (the second Figure S2 - there are two Figure S2s) has been relegated to the SI.

*We moved Figures 4, 10, and 16 to the supplement. That still leaves 14 figures, which we recognize is high, but we feel that all of those that remain are important for accompanying the text.*

- The abstract is also too long. It's not immediately clear what "transmission efficiencies" are upon reading the abstract without first reading the manuscript.

*We addressed both of these issues by deleting the following from the abstract: "The transmission efficiency of $O_3$ and $CO_2$ for all RH and of $SO_2$ for low RH exceeds 90 %, while it falls to about 70 % for $SO_2$ at 100 % RH. Particle transmission efficiency increases with increasing particle diameter from 0.67 for 0.050 $\mu m$ particles to 0.98 at 0.20 $\mu m$, while that of the ~3.3 $\mu m$ droplets formed on seed particles is greater than 80 %."*

- Why is K2SO4 used instead of other particle types which may be more representative of atmospheric aerosol or more typical of laboratory studies of aerosol and cloud chemistry? I think the question is answered several pages later on line 496 but should also be acknowledged when this is first mentioned. Also note that "neutral" isn't really representative of ambient aerosol or cloudwater pH (cf. Pye et al. 2020)

*It may be that the reviewer didn't notice the description accompanying the initial mention of $K_2SO_4$ starting on Line 210: "To date, most experiments have used potassium sulfate ($K_2SO_4$) seed particles because they are non-acidic, have a dynamic shape factor close to 1, and effloresce at an RH of about 60 % (Freney et al., 2009), which permits measurements without ALW at higher RH than would be possible with common aerosol types such as ammonium sulfate and sodium chloride."*

- the size dependence of the particle transmission efficiency is mentioned, and compared with other studies, but no physical explanation for decreasing transmission for smaller particles is given. One would expect the opposite.

*The original sentence: "The particle transmission efficiency increases with increasing particle size, from 0.67 for 0.050 μm particles, to 0.94 at 0.080 μm, and 0.98 at 0.20 μm." was extended to (briefly) explain the size dependence: "As expected for the particle size range considered for which the dominant loss mechanisms are Brownian motion and electrostatic attraction to charged surfaces, the particle transmission efficiency increases with increasing particle size, from 0.67 for 0.050 μm particles, to 0.94 at 0.080 μm, and 0.98 at 0.20 μm."*

- The SO2 transmission efficiency at high RH is low. Presumably studying SO2 multiphase chemistry was one of the main intended purposes of this apparatus. Can the authors comment on how this issue may impact the design of future studies?

*The experiments described in the manuscript in which $SO_2$ was injected and conversion to sulfate aerosol measured were meant to test the reactor with an extensively studied chemical system. The primary utility of the APPA is not for studying reaction of soluble gases that are directly injected, but rather further reaction of soluble products of gas phase oxidation of injected insoluble gases. Wall losses of those soluble products are expected to be lower simply because they will encounter a cloud of droplets in their path between where they form and the wall.*

- line 444: 'described in by (Mitroo et al)'... clean that up

*Now "described in Mitroo et al. (2018)."*

- residence time distribution: why is there a distribution at all if the gases or particles are introduced in the same location and sampled in the same location (if this is not the case it's not possible to tell from the text)? How are three measurements sufficient to construct the whole RTD curve and detect the absence of a long tail?

*Some smearing of an input pulse of particles or gases is unavoidable because the laminar velocity profile through the reactor is parabolic and not flat. The RTD tests are intended primarily to evaluate the contribution of things like mixing at the entrance and exit. The particle and gas measurements made at the outlet of the reactor were continuous and high time resolution, spanning the time between injection and several minutes later when concentrations had decreased to ~0. The "three" measurements were simply repeated tests of the same thing to ensure that the results were repeatable.*

- line 490 - I would either delete or rephrase this. You are presumably introducing this apparatus for the first time in this manuscript, why are you making this statement about it not being applied very often to cloud chemistry studies when that is the most novel and unique application?

*The sentence in question is "Though, like most OFRs, the APPA reactor is most often used to measure aerosol formation resulting from OH chemistry, conditions inside the reactor during cloud chemistry 490 experiments were first evaluated through the well-studied oxidation of dissolved $SO_2$ by $O_3$, leading to formation of sulfuric acid and growth of the particles on which the droplets formed."*

*The intent of the sentence is not to suggest that the reactor will not often be used to study cloud chemistry, but rather that it will not often be used to study aqueous phase oxidation by ozone and not hydroxyl radical.*

- Figure 9 and Figure S2. See my comment above about the second Figure S2.  It is much more interesting than Figure 9.  Figure 9 has been sufficiently described in the text lines 500-507. It could be made panel B of a figure which focuses on the data in Figure S2, moved to SI, or eliminated.  How was the low SO2 transmission efficiency dealt with in these calculations?

*We wanted to include one figure that shows the change in particle size distribution resulting from addition of secondary aerosol.  We felt that original Figure 14 (Figure 12 in revision) was a better choice because it shows the result of the type of experiment the APPA is primarily designed for.  As noted above, the $SO_2$ to $SO_4$ conversion experiment was meant to test the reactor with a well-studied chemical system and, thus, it seems more important to include results that can help readers visually evaluate how similar the experimental results are to expectations.  The figure in the supplement shows how the size distribution is altered, but does not allow a reader to evaluate the meaning of the extent of change.*

*The $SO_2$ concentration used in the calculations was determined based on a known concentration in gas mixture and a measured flow rate.  No correction for wall loss was included.  Though it may seem that this would result in a significant low bias based on original Figure 6, those transmission efficiency tests were done without introduced droplets.  Thus, much of the $SO_2$ lost during the transmission efficiency tests will instead dissolve and react in the droplets.*

- Is the relatively short residence time of APPA compared to a chamber experiment an issue when it comes to SOA studies?

*Absolutely, but this is an issue that affects all OFRs and is discussed at length in other publications. Adding a discussion here would add even more to an already long manuscript.*

- What is the reader supposed to take away from the SOA yield studies shown in Figure 15? Were these yields consistent with expectations?

*The primary reason for including the discussion and figure was to show the clear influence of liquid water on the amount of secondary aerosol formed.*

*The following was added to provide some context for the measured yields:  "The yield measured with no liquid water present is similar to the 0.18 – 0.28 range reported by Nakao et al. (2011), though the comparison is indirect because of differences in OHexp and organic mass loading between the two studies."*

- funding acknowledgement seems to be missing.

*The development of the APPA was supported by internal university funding sources and not a funding agency.*

---

## Author Comment (AC2)

Xu et al. present the evaluation of their "Accelerated Production and Processing of Aerosols" (APPA) OFR that is designed to enable aqueous-phase OH-initiated oxidative aging processes. The APPA OFR combines the authors' PFA OFR with injected K2SO4 seed particles at the inlet and controlled humidification that enables control of the particle liquid water content (the range of RH discussed in this paper is 40 - 100% RH). The authors perform characterization studies of gas and particle transmission efficiencies, residence time distributions, droplet size distributions, and radiation/oxidant profiles. To demonstrate applications of the APPA OFR, they generated sulfuric acid from SO2/O3 reactions, benzene OH-SOA in the presence of dry/aqueous seeds and "cloud droplets", and aged ambient aerosol.

Comments

While the APPA OFR appears to be capable of generating sulfuric acid from the aqueous-phase SO2/O3 reaction, its ability to initiate aqueous-phase OH oxidation chemistry was not conclusively demonstrated here. The authors did investigate SOA generated from gas-phase OH oxidation of benzene, followed by partitioning of OVOC/SOA into ALW/droplets. While this is a novel application that the APPA OFR seems to be well suited to, this is not a demonstration of aqSOA formation according to even the authors' own definition: "water-soluble products of gas-phase chemistry [that] enter cloud droplets or aerosol liquid water and react in the aqueous phase with the hydroxyl radical (OH) or other oxidants" (L83-L86). In that regard, I think they should have used K2SO4 seed particles containing H2O2 and/or H2O2/FeSO4 to initiate aqSOA formation (e.g. Nguyen et al., 2013; Daumit et al., 2016), then repeated the same experiments without H2O2/FeSO4, and inferred the difference in aerosol loading and composition as aqSOA. It is not clear to me why this was not done here – this is what I would need to see to be convinced that the APPA OFR can be used to investigate aqSOA formation.

*We certainly recognize that it is challenging to isolate the contribution of aqueous phase chemistry in a system in which it is occurring simultaneously with gas phase oxidation. But that is exactly the type of chemical system the APPA was developed to study. It is true that water soluble products of gas phase oxidation will preferentially partition into the aqueous aerosol and droplets, but without further oxidation those species will evaporate with the water as the aerosol/droplets exit the reactor, pass through two Nafion driers, and travel through a few meters of stainless tubing prior to measurement in the SMPS or AMS. Though the precursor studied was different (isoprene), Lamkaddam et al. (2021) reported that the aqSOA formed in their wetted-wall flow reactor could not be explained as simply the result of dissolution of products of gas phase oxidation: "…the dissolution of isoprene oxidized vapors cannot explain the aqSOA production and provides strong evidence that aqSOA results from OH reactions of the dissolved species in the aqueous phase."*

*Adding $H_2O_2$ and/or $H_2O_2/FeSO_4$ would be challenging for this system. As is described, the seed particles/droplets are always produced through size classification of an atomized and dried aerosol. $H_2O_2$ from the atomizer solution would be lost to evaporation unless the aerosol were not dried below the ERH of $K_2SO_4$ of approximately 65%. Uncertainty in the calculated change in mass concentration of*

*the aerosol following processing in the reactor would be substantially higher if a correction for the water content of the aerosol when initially classified is required.  The alternative would be injection of gas-phase $H_2O_2$ into the reactor, but then estimating how much gets into the droplets and how much is lost to the walls would be difficult.  Furthermore, even with $H_2O_2/FeSO_4$ in the aerosol/droplets, it is likely that the source of OH near their surface where more reactive species would be oxidized would still be that from the gas phase (which is needed to generate the oxidation products) and so interpretation of the with/without $H_2O_2/FeSO_4$ would be difficult.*

Similarly, how did the authors conclude that aqueous phase OH oxidation was responsible for the increase in ambient OA oxidation state as RH was increased from 40%-->85%->100% (Fig. 17 and related text)? Hypothetically, couldn't this change have been driven by the higher RH (and LWC) promoting more efficient partitioning of low-volatility gas-phase oxidation products into the aerosol? For the reasons mentioned in above comment, it is not clear to me that this evolution in OA oxidation state was in fact due to aqueous phase oxidative aging in the APPA OFR.

*Similar to part of the response to the above comment, without subsequent oxidation in the aqueous phase, water-soluble products of the gas phase oxidation are expected to evaporate when the aerosol/droplets are dried prior to measurement in the SMPS and AMS.  It is also unclear why preferential partitioning to the aqueous aerosol/droplets would be more pronounced for more oxidized (and presumably lower volatility) species.*

Unless signification dilution flow is added downstream of the APPA OFR, its relatively low 1.5 L min-1 sample flow capacity limits its application outside of measurements that can be made with particle counter(s) and instruments such as an AMS. What design changes would need to be made to increase this flow capacity to something in the range of 5-10 L min-1 that is closer to other commonly used OFR techniques?

*We do not intend to redesign the reactor to accommodate higher flow rate instruments.  The reactor volume cannot easily be increased by increasing the length because of height restrictions in the lab and any increase achieved through increasing the diameter would likely result in broadening of the RTD.  The flow rate could easily be ~doubled to 3 L min$^{-1}$ if, as with reactors such as the PAM, a single outlet flow was used rather than subsampling just the core flow, but again at the expense of the RTD.*

L25 and L230 - Quantify "low RH"

*The sentence in which it was included on L25 was removed in response to a comment by Reviewer # 1 that the abstract was too long.*

*We could add (40%) in parentheses after "low RH", but the exact value is unimportant and what matters is that it is low enough that there is no ALW (which is stated in the sentence).*

L70 – Clarify which "new pathways" are being referred to here

*We were referring to those described in studies such as Liu et al. (2020), but should have made the connection between that sentence and the following one clearer. We added "As just one example," to the second sentence in the paragraph ("Liu et al. (2020 measured sulfate…).*

L160 – Rather than "The APPA…is typically operated as a 254 nm-type OFR", I suggest instead saying "the APPA reactor is typically operated in OFR254 mode"

*We changed the text to "When used as an OFR, the APPA reactor is operated in OFR254 mode," The suggested replacement could give the impression that it is sometimes operated in OFR185 mode (which it isn't).*

L376 – typo ("to a to a")

*Thank you. Fixed.*

L446-L448 – Please indicate the relative humidity that was established in the APPA OFR when the RTD measurements were conducted, and please clarify if the lamps were on or off. Is there any humidity-dependence to the RTD?

*The sentence "For both tests the RH was controlled to 40% and the UV lights were off." was added after the sentence "The particle and $CO_2$ concentrations in the outlet-center flow were measured with the CPC and $CO_2$ analyzers identified in the previous section."*

*Though we did not repeat the experiments at different RH, we do not believe that there should be a significant dependence for either particles or low solubility gases such as $CO_2$.*

L450 – Why are the gas/particle RTD's in the APPA OFR narrower than in the PFA OFR when the two reactors are nominally the same design?

*The biggest differences between the two reactors, and those we believe are responsible for the narrower RTD with the APPA are: i) an inlet that was redesigned to minimize development of a jet near the core of the reactor and ii) addition of a water jacket around the reactor that minimizes temperature gradients and the convective mixing they would promote.*

L464 – I did not notice any explicit discussion of temperature control in the APPA OFR in this section.

*Thank you. The section title has been changed to just "Droplet size distribution"*

L492 - The experiments describing sulfuric acid formation from SO2/O3 were not clearly described. I assume sulfuric acid was generated from SO2 + O3 --> SO3 + O2 followed by SO3 + H2O --> H2SO4, but it would be useful to clarify this. How is the concentration of "dissolved" SO2 controlled and measured? Is O3 uptake onto the K2SO4 seed particles required to initiate this reaction?

*The original introduction to this section included:  "...first evaluated through the well-studied oxidation of dissolved SO$_2$ by O$_3$, leading to formation of sulfuric acid and growth of the particles on which the droplets formed."*

*Which has now been expanded to "...first evaluated through the well-studied oxidation of dissolved SO$_2$ by O$_3$, leading to growth of the particles on which the droplets formed accompanying the formation of sulfuric acid through the generic reaction:*

$$S(IV) + O_3 \rightarrow S(VI) + O_2$$

*Where S(IV) represents the +4 oxidation state sulfur species $SO_2 \cdot H_2O$, $HSO_3^-$, and $SO_3^{2-}$ that will not remain in the aerosol phase following evaporation of the droplet and S(VI) represents the +6 oxidation state sulfur species $H_2SO_4$, $HSO_4^-$, and $SO_4^{2-}$ that will remain in the aerosol phase."*

*Regarding the amount of SO$_2$ and O$_3$ in the droplets, these are controlled only by the gas phase concentrations, the Henry's Law constants, and, for SO$_2$, the dissociation equilibrium coefficients.  Those parameters and the reaction rate constants were used to calculate the expected increase in diameter shown in original Figure 9 (not Figure 8).  Rather than including all of the details of those calculations, we referenced Caffrey et al. (2001), which includes a thorough description of the relevant chemistry and equations.*

L583 – Typo (the)

*We inserted "the" in front of GC-FID.*

L608- Assuming that the authors are referring to the benzene/OH system here, I disagree that the "distribution of [benzene] oxidation products and their OH reaction constant(s) are generally unknown."See, for example, Xu et al. (2020); Priestley et al. (2021).

*You are correct.  This statement is reasonable for some precursors, but not for benzene.  We have removed the sentence.*

L721. Please clarify the author contributions of C. Le and D. R. Cocker.

*The author contributions has been updated to include C. Le and D. R. Cocker.  "D. R. Collins designed the reactor and edited the paper. N. Xu performed the experiments and simulations, processed the data, and wrote the paper.  C. Le and D. R. Cocker contributed to some of the experiments and data analysis."*

Some of the figures should be moved to the Supplement - in my opinion, Figures 4, 8, 9, 10, 12, 13, and 16 would be a better fit there.

*We moved Figures 4, 10, and 16 to the Supplement. We feel that the others are needed for understanding the accompanying text in the manuscript and should be left there so that readers can view them more easily.*

The KinSim mechanism and case files that were used here should be uploaded with the Supplement.

*These will be included in the Supplement.*

References

T. B. Nguyen, M. M. Coggon, R. C. Flagan, and J. H. Seinfeld, Reactive Uptake and Photo-Fenton Oxidation of Glycolaldehyde in Aerosol Liquid Water. Environmental Science & Technology 2013 47 (9), 4307-4316. DOI: 10.1021/es400538j

Kelly E. Daumit, Anthony J. Carrasquillo, Rebecca A. Sugrue, and Jesse H. Kroll . Effects of Condensed-Phase Oxidants on Secondary Organic Aerosol Formation. The Journal of Physical Chemistry A 2016, 120 (9) , 1386-1394. https://doi.org/10.1021/acs.jpca.5b06160

Lu Xu, Kristian H. Møller, John D. Crounse, Henrik G. Kjaergaard, and Paul O. Wennberg, New Insights into the Radical Chemistry and Product Distribution in the OH-Initiated Oxidation of Benzene, Environmental Science & Technology 2020 54 (21), 13467-13477. DOI: 10.1021/acs.est.0c04780.

Priestley, M., Bannan, T. J., Le Breton, M., Worrall, S. D., Kang, S., Pullinen, I., Schmitt, S., Tillmann, R., Kleist, E., Zhao, D., Wildt, J., Garmash, O., Mehra, A., Bacak, A., Shallcross, D. E., Kiendler-Scharr, A., Hallquist, Å. M., Ehn, M., Coe, H., Percival, C. J., Hallquist, M., Mentel, T. F., and McFiggans, G.: Chemical characterisation of benzene oxidation products under high- and low-NOx conditions using chemical ionisation mass spectrometry, Atmos. Chem. Phys., 21, 3473–3490, https://doi.org/10.5194/acp-21-3473-2021, 2021.

---

## Referee Report (RR1)

While the APPA OFR appears to be capable of generating sulfuric acid from the aqueous-phase SO2/O3 reaction, its ability to initiate aqueous-phase OH oxidation chemistry was not conclusively demonstrated here. The authors did investigate SOA generated from gas-phase OH oxidation of benzene, followed by partitioning of OVOC/SOA into ALW/droplets. While this is a novel application that the APPA OFR seems to be well suited to, this is not a demonstration of aqSOA formation according to even the authors' own definition: "water-soluble products of gas-phase chemistry [that] enter cloud droplets or aerosol liquid water and react in the aqueous phase with the hydroxyl radical (OH) or other oxidants" (L83-L86). In that regard, I think they should have used K2SO4 seed particles containing H2O2 and/or H2O2/FeSO4 to initiate aqSOA formation (e.g. Nguyen et al., 2013; Daumit et al., 2016), then repeated the same experiments without H2O2/FeSO4, and inferred the difference in aerosol loading and composition as aqSOA. It is not clear to me why this was not done here – this is what I would need to see to be convinced that the APPA OFR can be used to investigate aqSOA formation.

*We certainly recognize that it is challenging to isolate the contribution of aqueous phase chemistry in a system in which it is occurring simultaneously with gas phase oxidation. But that is exactly the type of chemical system the APPA was developed to study. It is true that water soluble products of gas phase oxidation will preferentially partition into the aqueous aerosol and droplets, but without further oxidation those species will evaporate with the water as the aerosol/droplets exit the reactor, pass through two Nafion driers, and travel through a few meters of stainless tubing prior to measurement in the SMPS or AMS. Though the precursor studied was different (isoprene), Lamkaddam et al. (2021) reported that the aqSOA formed in their wetted-wall flow reactor could not be explained as simply the result of dissolution of products of gas phase oxidation: "…the dissolution of isoprene oxidized vapors cannot explain the aqSOA production and provides strong evidence that aqSOA results from OH reactions of the dissolved species in the aqueous phase." Adding H2O2 and/or H2O2/FeSO4 would be challenging for this system. As is described, the seed particles/droplets are always produced through size classification of an atomized and dried aerosol. H2O2 from the atomizer solution would be lost to evaporation unless the aerosol were not dried below the ERH of K2SO4 of approximately 65%. Uncertainty in the calculated change in mass concentration of the aerosol following processing in the reactor would be substantially higher if a correction for the water content of the aerosol when initially classified is required. The alternative would be injection of gas‑phase H2O2 into the reactor, but then estimating how much gets into the droplets and how much is lost to the walls would be difficult. Furthermore, even with H2O2/FeSO4 in the aerosol/droplets, it is likely that the source of OH near their surface where more reactive species would be oxidized would still be that from the gas phase (which is needed to generate the oxidation products) and so interpretation of the with/without H2O2/FeSO4 would be difficult.*

My opinion is that conclusive demonstration of the APPA's ability to mimic aqueous phase OH processes was not shown in this manuscript. OH was generated in the gas phase and its reaction with benzene and its oxidation products also occurred in the gas phase. Given diffusional limitations of OH to the surface of the particles, it is not clear to me how significant aqueous phase OH oxidation could have occurred here.

Analogous to the aqueous $SO_2/O_3$ reaction presented by the authors, $H_2O_2$ or $H_2SO4/FeSO_4$ are, to my knowledge, the canonical precursors for initiating controlled aqueous phase OH oxidation in laboratory studies. In my opinion, to demonstrate the APPA OFR for this type of application, the experiments could have, and should have been designed to accommodate their use by working around their limitations. It is a fair point that drying/size selecting the aerosols, and/or interpreting results with/without $H_2SO_4/FeSO_4$, could make it challenging to implement them. However, there are other ways that these aqueous OH precursors could have been used: for example, seed aerosols could have been maintained above the ERH of $K_2SO_4$, or polydisperse seeds could have been used, and aerosol compositional changes with the AMS could have been measured with and without illumination of $H_2O_2/FeSO_4/K_2SO_4$ particles mixed with SOA surrogates or water-soluble SOA extracts rather than needing to generate the SOA in the gas phase as was done here.

In its current form, in my opinion, the revised paper adequately demonstrates the APPA's ability to study the aqueous phase $S(IV)/O_3$ reaction because the gas-phase $SO_2 + O_3$ reaction rate is negligible. It is hard for me to come up with any alternative explanation other than that S(VI) formation is occurring via aqueous phase $O_3$ oxidation processes. I don't think the same can be said for the benzene/OH reaction, so I think it is misleading to claim that the APPA can be used for this type of application.

Similarly, how did the authors conclude that aqueous phase OH oxidation was responsible for the increase in ambient OA oxidation state as RH was increased from 40%-->85%->100% (Fig. 17 and related text)? Hypothetically, couldn't this change have been driven by the higher RH (and LWC) promoting more efficient partitioning of low-volatility gas-phase oxidation products into the aerosol?

*For the reasons mentioned in above comment, it is not clear to me that this evolution in OA oxidation state was in fact due to aqueous phase oxidative aging in the APPA OFR. Similar to part of the response to the above comment, without subsequent oxidation in the aqueous phase, water-soluble products of the gas phase oxidation are expected to evaporate when the aerosol/droplets are dried prior to measurement in the SMPS and AMS. It is also unclear why preferential partitioning to the aqueous aerosol/droplets would be more pronounced for more oxidized (and presumably lower volatility) species.*

My current interpretation of the results shown here is that the yield of gas-phase benzene/OH oxidation products that partition into aerosol liquid water simply increased as a function of increasing humidity in the APPA, and that this is what is driving the humidity-dependent compositional changes (rather than aqueous-phase OH oxidation).

---

## Referee Report (RR2)

Overall, the FIGAERO-HR-ToF-CIMS measurements of toluene SOA that were added to the revised manuscript significantly strengthen proof of principle for application of the APPA to study aqSOA formation. The specific discussion of those results would benefit from some clarification text and additional analysis that I suggest below. Once these comments are addressed, I would support publication of the manuscript in AMT.

**Comments specific to FIGAERO-HR-ToF-CIMS data presentation and interpretation**

1. L638-641: I suggest rewriting this text for greater clarity/conciseness, something like: "Filter Inlet for Gases and Aerosols (FIGAERO) coupled to an iodide adduct High-Resolution Time-of-Flight Chemical Ionization Mass Spectrometer (FIGAERO-HR-ToF-CIMS; Aerodyne Research, Inc.)."

2. L676: Replace 'CIMS' with 'FIGAERO-HR-ToF-CIMS'

3. L677-684: I suggest rewriting these sentences for clarity/conciseness, something like: "The signal observed at m/z = 217 ($C_2H_2O_4I^-$) was absent from the mass spectrum obtained from SOA generated in the APPA at 40% RH (Fig. S7a) but was the dominant peak in spectra obtained from SOA generated at 85 and 100 % RH (Figs. S7b and c). The $C_2H_2O_4I^-$ signal corresponds to oxalic acid, which can be produced following reactive uptake, hydration, and multi-generational OH oxidation of glyoxal in the aqueous phase (Lim et al., 2010). While glyoxal is a major gas-phase OH oxidation product of toluene (Volkamer et al., 2001), oxalic acid is not generated via subsequent gas-phase OH oxidation of glyoxal (Warneck, 2003), which is why it is observed in Figs. S7b-c but not in Fig. S7a."

4. Figure S7 and related discussion.
   a. Add (a), (b), and (c) labels to the mass spectra obtained at 40%, 85%, and 100%, respectively.
   b. Add OH exposure values at which Figure S7a, b and c were obtained - this information is critical to properly interpret results in the context of the SOA yields presented in Fig. 12 - and reference the corresponding OH exposure values in the Sect. 3.2.1 discussion.
   c. Please extend the FIGAERO-HR-ToF-CIMS spectra to show signals down to m/z = 200, which would then include known aqueous phase glyoxal oxidation products at m/z = 201 ($C_2H_2O_3I^-$ , glyoxylic acid) and/or 203 ($C_2H_4O_3I^-$ , glycolic acid), both of which are detected with iodide adduct CIMS at even greater sensitivity than oxalic acid (e.g. Table S1 in Lee et al., 2014).
   d. Glyoxylic acid is a direct precursor to oxalic acid via aqueous phase OH oxidation (e.g. Lim et al., 2010). It would be especially noteworthy if the $C_2H_2O_3I^-$: $C_2H_2O_4I^-$ ratio is higher at 85% RH than 100% RH, for example. If glyoxylic/glycolic acid signals are not present, it

could be because they were already fully consumed at the OH exposures that were used, in which case that could be briefly discussed in the text.

e. Figure S7 shows the intensity of FIGAERO-HR-ToF-CIMS signals obtained from m/z = 210-360 normalized to $C_3H_2O_4I^-$ (Fig. S7a) or $C_2H_2O_4I^-$ (Figs. S7b-c). While this may be adequate for the sole purpose of showing that $C_2H_2O_4I^-$ is either present or not, it would be a much clearer presentation if the ion signals were be normalized to the calculated mass of SOA collected on the filters – which could be estimated from the SOA mass concentration measured with AMS or SMPS. This would then more closely relate the abundance of specific ions that were measured to the SOA yields that are plotted in Fig S12. For example, while the mass spectra of SOA generated at 85% RH and 100 % RH look similar, the SOA yield is up to a factor of 2 higher in the 100% RH case. Why is that - are the same products generated in both cases, in different yields? It's impossible to tell the way the data are currently presented.

**Other comments**

5. L595: for context, I suggest indicating that the $OHR_{ext}$ from 150-250 ppb of added $SO_2$ is approximately 0.5-0.8 $s^{-1}$.

6. L606-621 and Figure 10: This content should either be deleted or moved to the supplement – while interesting and showing that the authors' OH exposure calibration studies were done carefully, the discussion of this result/trend is not specific to the APPA and is more of a commentary on the use of OH tracers whose products can also react with OH.

7. Figure 6 is missing (a) and (b) labels.

8. In Figure 6(b), the "well-stirred" RTD trace looks black, whereas the legend trace indicates it should be light grey.

**References**

Lim, Y. B., Tan, Y., Perri, M. J., Seitzinger, S. P., and Turpin, B. J.: Aqueous chemistry and its role in secondary organic aerosol (SOA) formation, Atmos. Chem. Phys., 10, 10521–10539, https://doi.org/10.5194/acp-10-10521-2010, 2010.

Lee, Ben H., Lopez-Hilfiker, Felipe D., Mohr, Claudia, Kurtén, Theo, Worsnop, Douglas R., and Thornton, Joel A. An Iodide-Adduct High-Resolution Time-of-Flight Chemical-Ionization Mass Spectrometer: Application to Atmospheric Inorganic and Organic Compounds. Environ. Sci. Technol., 48, 11, 6309-6317, https://doi.org/10.1021/es500362a, 2014.

---

## Author Response (AR2)

**Response to Referee 1**

The authors' responses were not completely satisfactory. In several cases they simply repeated lines from the text instead of addressing the reviewers' concerns, or a response was given but no change was made to the manuscript to clarify the issue which was raised. As such, there were several missed opportunities to make the manuscript stronger in this round of revisions.

Even in cases where the authors think a comment was made based on a reviewer's misunderstanding of the text or the intent of the study, rather than just dismissing the comment, realize that there may be an issue with the clarity of the manuscript which the authors don't perceive, but the reviewers do, as outsiders to the project. Use these instances as opportunities to clarify the text to eliminate reader misunderstandings in the final version.

I will point out a few specific issues here but this is not the full extent of the missed opportunities for improving the manuscript. I suggest the authors read through the response and consider carefully where they could be more thorough.

- As previously stated by the reviewer, "neutral" or "non-acidic" is not an environmentally realistic property of atmospheric aerosols, and not always for cloud droplets. The authors need to include a statement in the manuscript to this effect rather than ignoring this comment.

*We added the following sentence (line 211): "The pH of pure $K_2SO_4$ aqueous particles or cloud droplets that form on them is close to 7, which is not representative of typical atmospheric aerosols, but simplifies interpretation of experiments for systems with significant pH dependence."*

- "OFR254 mode" is jargon that has no meaning outside the OFR community. If you wish to use this terminology it's important to also include a general definition.

*We revised that sentence. It is now: "When used as an OFR, the APPA reactor is operated with lamps producing only 254 nm UV (and not also 185 nm; often referred to as OFR254), with OH produced from photolysis of $O_3$ that is produced externally by an $O_3$ generator (Jelight Co., Inc., Model 610) and introduced into the reactor."*

- Both reviewers suggested moving several figures to the SI, with explicit suggestions for figure removal made by both reviewers. The authors disregarded most of these recommendations from both reviewers, keeping a total of 14 figures, which is still too many.

*We also moved Figure 10 to the Supplement, leaving 13 in the main manuscript now.*

**Response to Referee 2**

My responses to the author's replies are in red text below. – Referee #2
* * *
While the APPA OFR appears to be capable of generating sulfuric acid from the aqueous-phase SO2/O3 reaction, its ability to initiate aqueous-phase OH oxidation chemistry was not conclusively demonstrated here. The authors did investigate SOA generated from gas-phase OH oxidation of benzene, followed by partitioning of OVOC/SOA into ALW/droplets. While this is a novel application that the APPA OFR seems to be well suited to, this is not a demonstration of aqSOA formation according to even the authors' own definition: "water-soluble products of gas-phase chemistry [that] enter cloud droplets or aerosol liquid water and react in the aqueous phase with the hydroxyl radical (OH) or other oxidants" (L83-L86). In that

regard, I think they should have used K2SO4 seed particles containing H2O2 and/or H2O2/FeSO4 to initiate aqSOA formation (e.g. Nguyen et al., 2013; Daumit et al., 2016), then repeated the same experiments without H2O2/FeSO4, and inferred the difference in aerosol loading and composition as aqSOA. It is not clear to me why this was not done here – this is what I would need to see to be convinced that the APPA OFR can be used to investigate aqSOA formation.

*RESPONSE WITH FIRST REVISION: We certainly recognize that it is challenging to isolate the contribution of aqueous phase chemistry in a system in which it is occurring simultaneously with gas phase oxidation. But that is exactly the type of chemical system the APPA was developed to study. It is true that water soluble products of gas phase oxidation will preferentially partition into the aqueous aerosol and droplets, but without further oxidation those species will evaporate with the water as the aerosol/droplets exit the reactor, pass through two Nafion driers, and travel through a few meters of stainless tubing prior to measurement in the SMPS or AMS. Though the precursor studied was different (isoprene), Lamkaddam et al. (2021) reported that the aqSOA formed in their wetted-wall flow reactor could not be explained as simply the result of dissolution of products of gas phase oxidation: "…the dissolution of isoprene oxidized vapors cannot explain the aqSOA production and provides strong evidence that aqSOA results from OH reactions of the dissolved species in the aqueous phase." Adding $H_2O_2$ and/or $H_2O_2/FeSO_4$ would be challenging for this system. As is described, the seed particles/droplets are always produced through size classification of an atomized and dried aerosol. $H_2O_2$ from the atomizer solution would be lost to evaporation unless the aerosol were not dried below the ERH of $K_2SO_4$ of approximately 65%. Uncertainty in the calculated change in mass concentration of the aerosol following processing in the reactor would be substantially higher if a correction for the water content of the aerosol when initially classified is required. The alternative would be injection of gas-phase $H_2O_2$ into the reactor, but then estimating how much gets into the droplets and how much is lost to the walls would be difficult. Furthermore, even with $H_2O_2/FeSO_4$ in the aerosol/droplets, it is likely that the source of OH near their surface where more reactive species would be oxidized would still be that from the gas phase (which is needed to generate the oxidation products) and so interpretation of the with/without $H_2O_2/FeSO_4$ would be difficult.*

My opinion is that conclusive demonstration of the APPA's ability to mimic aqueous phase OH processes was not shown in this manuscript. OH was generated in the gas phase and its reaction with benzene and its oxidation products also occurred in the gas phase. Given diffusional limitations of OH to the surface of the particles, it is not clear to me how significant aqueous phase OH oxidation could have occurred here.

Analogous to the aqueous $SO_2/O_3$ reaction presented by the authors, $H_2O_2$ or $H_2SO4/FeSO_4$ are, to my knowledge, the canonical precursors for initiating controlled aqueous phase OH oxidation in laboratory studies. In my opinion, to demonstrate the APPA OFR for this type of application, the experiments could have, and should have been designed to accommodate their use by working around their limitations. It is a fair point that drying/size selecting the aerosols, and/or interpreting results with/without $H_2SO_4/FeSO_4$, could make it challenging to implement them. However, there are other ways that these aqueous OH precursors could have been used: for example, seed aerosols could have been maintained above the ERH of $K_2SO_4$, or polydisperse seeds could have been used, and aerosol compositional changes with the AMS could have been measured with and without illumination of $H_2O_2/FeSO_4/K_2SO_4$ particles mixed with SOA surrogates or water-soluble SOA extracts rather than needing to generate the SOA in the gas phase as was done here.

In its current form, in my opinion, the revised paper adequately demonstrates the APPA's ability to study the aqueous phase $S(IV)/O_3$ reaction because the gas-phase $SO_2 + O_3$ reaction rate is negligible. It is hard for me to come up with any alternative explanation other than that S(VI) formation is occurring via aqueous phase $O_3$ oxidation processes. I don't think the same can be said for the benzene/OH reaction, so I think it is misleading to claim that the APPA can be used for this type of application.

Similarly, how did the authors conclude that aqueous phase OH oxidation was responsible for the increase in ambient OA oxidation state as RH was increased from 40%-->85%->100% (Fig. 17 and related text)? Hypothetically, couldn't this change have been driven by the higher RH (and LWC) promoting more efficient partitioning of low-volatility gas-phase oxidation products into the aerosol?

For the reasons mentioned in above comment, it is not clear to me that this evolution in OA oxidation state was in fact due to aqueous phase oxidative aging in the APPA OFR. Similar to part of the response to the above comment, without subsequent oxidation in the aqueous phase, water-soluble products of the gas phase oxidation are expected to evaporate when the aerosol/droplets are dried prior to measurement in the SMPS and AMS. It is also unclear why preferential partitioning to the aqueous aerosol/droplets would be more pronounced for more oxidized (and presumably lower volatility) species.

My current interpretation of the results shown here is that the yield of gas-phase benzene/OH oxidation products that partition into aerosol liquid water simply increased as a function of increasing humidity in the APPA, and that this is what is driving the humidity-dependent compositional changes (rather than aqueous-phase OH oxidation).

*We recognized that our assertion that much of the SOA was forming through aqueous phase reactions was not adequate. Reviews are always valuable in highlighting assumptions that are trusted, but not verified. We have made significant changes to the manuscript that we believe demonstrate a significant role of aqueous phase chemistry. Specifically, we replaced the description of SOA formed from oxidation of benzene with that of toluene. Importantly, we collected the aerosol at the exit of the reactor for each RH condition and analyzed it using a FIGAERO-CIMS. Most of the results included in the manuscript for toluene SOA are similar to those for benzene SOA. The rationale for making the*

*change is the CIMS analysis results now included in Figure S7 in the supplement (copied below), which show oxalic acid makes up a substantial fraction of the aerosol formed when aqueous aerosol and cloud droplets were present, but a negligible amount when only dry seed aerosol was. The paragraph added to the text in support of the role of aqueous phase chemistry is:*

*"The observed increase in SOA yield with increasing liquid water content is believed to result from further oxidation in the aqueous phase of the products of the gas phase oxidation of toluene. An alternative explanation that must be considered is that the enhancement is simply a consequence of the increased surface area available for condensation of low volatility products of the gas phase oxidation. Support for a significant role of aqueous phase chemistry comes from the aerosol composition measured with the CIMS. Mass spectra of the SOA collected for the three RH (and liquid water) conditions are shown in Fig. S7. The spectra obtained for the aerosol collected during the 85 % RH and 100 % RH experiments have a dominant peak at the m/z of oxalic acid (217 with I⁻) that is more than 5 times higher than the next highest peak, while it is absent in the spectrum obtained for the aerosol from the 40 % RH experiment. Oxalic acid is not produced from gas phase chemistry (Warneck, 2003), but has been shown to be an important product of oxidation of aromatics including toluene in the presence of ALW or cloud water. Specifically, oxalic acid is produced from aqueous phase oxidation of glyoxal, which is formed from the gas-phase oxidation of toluene with a yield of as high as ~ 0.4 (Volkamer et al., 2001)."*

[Figure]

**Figure S7.** Mass spectra of SOA produced from OH-oxidation of toluene in the presence of dry seed particles (40 % RH), aqueous seed particles (85 % RH), and cloud droplets (100 % RH). The aerosol was collected on PTFE membrane filters and then evaporated from a FIGAERO inlet connected to an HR-ToF-CIMS. A peak corresponding to oxalic acid ($C_2H_2O_4$) was dominant for both the 85 % RH and 100 % RH cases but absent for the 40 % RH case (note that the amplitudes of the oxalic acid peaks in both the 85 % and 100 % RH spectra were multiplied by 0.2).

---

## Author Response (AR3)

**Response to Referee 1**

Overall, the FIGAERO-HR-ToF-CIMS measurements of toluene SOA that were added to the revised manuscript significantly strengthen proof of principle for application of the APPA to study aqSOA formation. The specific discussion of those results would benefit from some clarification text and additional analysis that I suggest below. Once these comments are addressed, I would support publication of the manuscript in AMT.

We are very thankful for the recommendations of changes and additions provided by the referee.

**Comments specific to FIGAERO-HR-ToF-CIMS data presentation and interpretation**

1. L638-641: I suggest rewriting this text for greater clarity/conciseness, something like: "Filter Inlet for Gases and Aerosols (FIGAERO) coupled to an iodide adduct High-Resolution Time-of-Flight Chemical Ionization Mass Spectrometer (FIGAERO-HR-ToF-CIMS; Aerodyne Research, Inc.)."

   The sentence was revised as suggested.

2. L676: Replace 'CIMS' with 'FIGAERO-HR-ToF-CIMS'

   The acronym was replaced as suggested.

3. L677-684: I suggest rewriting these sentences for clarity/conciseness, something like: "The signal observed at m/z = 217 ($C_2H_2O_4I^-$) was absent from the mass spectrum obtained from SOA generated in the APPA at 40% RH (Fig. S7a) but was the dominant peak in spectra obtained from SOA generated at 85 and 100 % RH (Figs. S7b and c). The $C_2H_2O_4I^-$ signal corresponds to oxalic acid, which can be produced following reactive uptake, hydration, and multi-generational OH oxidation of glyoxal in the aqueous phase (Lim et al., 2010). While glyoxal is a major gas-phase OH oxidation product of toluene (Volkamer et al., 2001), oxalic acid is not generated via subsequent gas-phase OH oxidation of glyoxal (Warneck, 2003), which is why it is observed in Figs. S7b-c but not in Fig. S7a."

   The sentences were revised as suggested.

4. Figure S7 and related discussion.
   a. Add (a), (b), and (c) labels to the mass spectra obtained at 40%, 85%, and 100%, respectively.

      The labels were added.

b. Add OH exposure values at which Figure S7a, b and c were obtained - this information is critical to properly interpret results in the context of the SOA yields presented in Fig. 12 - and reference the corresponding OH exposure values in the Sect. 3.2.1 discussion.

The first sentence in the Figure S7 caption is now:

"Mass spectra of SOA produced from OH-oxidation of toluene in the presence of dry seed particles (40 % RH), aqueous seed particles (85 % RH), and cloud droplets (100 % RH), with OH exposures calculated from KinSim of $1.06 \times 10^{12}$, $1.03 \times 10^{12}$, and $1.01 \times 10^{12}$ molec. cm$^{-3}$ s, respectively.

c. Please extend the FIGAERO-HR-ToF-CIMS spectra to show signals down to m/z = 200, which would then include known aqueous phase glyoxal oxidation products at m/z = 201 ($C_2H_2O_3I^-$, glyoxylic acid) and/or 203 ($C_2H_4O_3I^-$, glycolic acid), both of which are detected with iodide adduct CIMS at even greater sensitivity than oxalic acid (e.g. Table S1 in Lee et al., 2014).

The x-axes were extended down to m/z = 200. As you will see, no additional peaks are evident.

d. Glyoxylic acid is a direct precursor to oxalic acid via aqueous phase OH oxidation (e.g. Lim et al., 2010). It would be especially noteworthy if the $C_2H_2O_3I^-$ : $C_2H_2O_4I^-$ ratio is higher at 85% RH than 100% RH, for example. If glyoxylic/glycolic acid signals are not present, it could be because they were already fully consumed at the OH exposures that were used, in which case that could be briefly discussed in the text.

As noted above, peaks corresponding to glycolic and glyoxylic acid are not evident in the extended spectra. The following was added to the text to explain this observation:

"Peaks are not evident in the spectra corresponding to glyoxylic acid ($C_2H_2O_3I^-$; m/z = 201) and glycolic acid ($C_2H_4O_3I^-$; m/z = 203), both of which are oxidation products of glyoxal and the former of which is a direct precursor to oxalic acid through aqueous phase oxidation (e.g., Lim et al., 2010). The absence of those compounds together with the high concentration of oxalic acid is attributed to a combination of their oxidation to near completion during the ~8-day equivalent photochemical aging in the reactor and to their evaporation from the residual particles following evaporation of the water from the droplets because of their substantially higher vapor pressures (~1 mmHg, ~0.02 mmHg, and 0.0002 mmHg for glyoxylic, glycolic, and oxalic acid, respectively; Brown, 2008).

e. Figure S7 shows the intensity of FIGAERO-HR-ToF-CIMS signals obtained from m/z = 210- 360 normalized to $C_3H_2O_4I^-$ (Fig. S7a) or $C_2H_2O_4I^-$ (Figs. S7b-c). While this may be adequate for the sole purpose of showing that $C_2H_2O_4I^-$ is either present or not, it would be a much clearer presentation if the ion signals were be normalized to the calculated mass of SOA collected on the filters – which could be estimated from the SOA mass concentration measured with AMS or SMPS. This would then more closely relate the abundance of specific ions that were measured to the SOA yields that are plotted in Fig S12. For example, while the mass spectra of SOA generated at 85% RH and 100 % RH look similar, the SOA yield is up to a factor of 2 higher in the 100% RH case. Why is that - are the same products generated in both cases, in different yields? It's impossible to tell the way the data are currently presented.

The spectra are now normalized by the collected SOA mass.

**Other comments**

5. L595: for context, I suggest indicating that the $OHR_{ext}$ from 150-250 ppb of added $SO_2$ is approximately 0.5-0.8 $s^{-1}$.

I calculate a range of 4.7 – 7.8 $s^{-1}$. Below is the approximate calculation for 150 ppb.

$$OHR_{ext} = (1.3 \times 10^{-12}\ cm^3\ s^{-1})(150 \times 10^{-9})(2.4 \times 10^{19}\ cm^{-3}) = 4.7\ s^{-1}$$

Nevertheless, that range is included in the modified sentence:

"For the simulations resulting in the values along the upper (black) curve, the only source of "external" OH reactivity ($OH_{ext}$) (Peng et al., 2015) was the ~ 4.7 to 7.8 $s^{-1}$ corresponding to the added 150 to 250 ppb $SO_2$."

6. L606-621 and Figure 10: This content should either be deleted or moved to the supplement – while interesting and showing that the authors' OH exposure calibration studies were done carefully, the discussion of this result/trend is not specific to the APPA and is more of a commentary on the use of OH tracers whose products can also react with OH.

The paragraph and figure were moved to the supplement. The following sentence was added to the end of the previous paragraph:

"A summary of the measurements and simulations for high RH experiments using benzene is provided in Supplement 5 and Fig. S7."

7. Figure 6 is missing (a) and (b) labels.

They were actually in there, but positioned above the graphs (and not very easy to see). They are now moved inside the graphs.

8. In Figure 6(b), the "well-stirred" RTD trace looks black, whereas the legend trace indicates it should be light grey.

   The colors are now correct.

**References**

Lim, Y. B., Tan, Y., Perri, M. J., Seitzinger, S. P., and Turpin, B. J.: Aqueous chemistry and its role in secondary organic aerosol (SOA) formation, Atmos. Chem. Phys., 10, 10521–10539, https://doi.org/10.5194/acp-10-10521-2010, 2010.

Lee, Ben H., Lopez-Hilfiker, Felipe D., Mohr, Claudia, Kurtén, Theo, Worsnop, Douglas R., and Thornton, Joel A. An Iodide-Adduct High-Resolution Time-of-Flight Chemical-Ionization Mass Spectrometer: Application to Atmospheric Inorganic and Organic Compounds. Environ. Sci. Technol., 48, 11, 6309-6317, https://doi.org/10.1021/es500362a, 2014.

The Lim et al. reference was already included.  We did not include the Lee et al. reference because it did not seem necessary to comment on the relative ionization efficiencies when no peaks corresponding to glycolic and glyoxylic acid are evident in the spectra.